# Distinct Element geomechanical modelling of the formation of sinkhole clusters within large-scale karstic depressions

Djamil Al-Halbouni[1,2], Eoghan P. Holohan[3], Abbas Taheri[4], Robert A. Watson[3], Ulrich Polom[5], Martin P.J. Schöpfer[6], Sacha Emam[7], and Torsten Dahm[1,2]

[1]Helmholtz Centre - German Research Centre for Geosciences (GFZ), Physics of Earthquakes and Volcanoes, Telegrafenberg, Potsdam 14473, Germany.
[2]University of Potsdam, Institute of Geosciences, P.O. Box 601553, Potsdam-Golm 14415, Germany
[3]UCD School of Earth Sciences, University College Dublin, Belfield, Dublin 4, Ireland.
[4]School of Civil, Environmental and Mining Engineering, University of Adelaide, Adelaide, South Australia 5005, Australia.
[5]Leibnitz Institute for Applied Geophysics (LIAG), Stilleweg 2, 30655 Hannover, Germany
[6]Department for Geodynamics and Sedimentology, University of Vienna, Athanstrasse 14, A-1090, Vienna, Austria
[7]Geomechanics and Software Engineer, Itasca Consultants S.A.S, Écully, France

*Correspondence to*: Djamil Al-Halbouni (halbouni.etal@posteo.de)

**Abstract**

The 2D Distinct Element Method (DEM) code (PFC2D_V5) is here used to simulate the evolution of subsidence-related karst landforms, such as single and clustered sinkholes, and associated larger-scale depressions. Subsurface material in the DEM model is removed progressively to produce an array of cavities; this simulates a network of subsurface groundwater conduits growing by chemical/mechanical erosion. The growth of the cavity array is coupled mechanically to the gravitationally-loaded surroundings, such that cavities can grow also in part by material failure at their margins, which in the limit can produce individual collapse sinkholes. Two end-member growth scenarios of the cavity array and their impact on surface subsidence were examined in the models: (1) cavity growth at the same depth level and growth rate; (2) cavity growth at progressively deepening levels with varying growth rates. These growth scenarios are characterised by differing stress patterns across the cavity array and its overburden, which are in turn an important factor for the formation of sinkholes and uvala-like depressions. For growth scenario (1), a stable compression arch is established around the entire cavity array, hindering sinkhole collapse into individual cavities and favouring block-wise, relatively even subsidence across the whole cavity array. In contrast, for growth scenario (2), the stress system is more heterogeneous, such that local stress concentrations exist around individual cavities leading to stress interactions and local wall/overburden fractures. Consequently, sinkhole collapses occur into individual cavities, which results in uneven, differential subsidence within a larger scale depression. Depending on material properties of the cavity-hosting material and the overburden, the larger-scale depression forms either by sinkhole coalescence or by wide-spread subsidence linked geometrically to the entire cavity array. The results from models with growth scenario (2), are in close agreement with surface morphological and subsurface geophysical observations from an evaporite karst area on the eastern shore of the Dead Sea.

## 1    Introduction

Karstification occurs worldwide in rocks like limestone, dolomite, gypsum, anhydrite and salt primarily by chemical dissolution (BGR et al., 2017). While subsurface-solution based drainage networks and connected void spaces resulting from karstification are hydrologically important for groundwater provision (Chen et al., 2017), such features reduce the mechanical stability of the geologic material and so may pose a significant hazard to humans and infrastructure. Sinkholes, also termed dolines, are the prominent karst landforms (Waltham et al., 2005). They form enclosed small- to large-scale depressions that are commonly considered to be morphological expressions of material removal in the underground and subsequent collapse of the overburden (Gutiérrez et al., 2014; De Waele et al., 2011; Waltham, 2016). Often, systems develop into agglomerations of closely spaced or coalesced dolines and elongated valley-like depressions, potentially revealing linear patterns of drainage (Waltham et al., 2005). Such sinkhole cluster development can be highly dynamic and partly accelerating, and may affect large areas in short times (e.g. Abelson et al., 2017). Understanding their development and where possible, their precursor signals occur is of utmost importance to mitigate their hazard and to promote sustainable land and water usage.

The main problem for unravelling the geometric and genetic relationships between sinkhole cluster development and larger-scale depressions in limestone karst areas, where such landforms have historically been best described, is that the landform evolution is controlled by the relatively slow dissolution kinetics of carbonate minerals. Consequently, the development of these karstic landform types is not directly observable in such areas, and furthermore, it is susceptible to long-term influences from climate change and tectonic activity. Indeed, the areas in which dolines and other karstic depressions have been historically best documented have been modified not only by karst processes but also by fluvial and/or glacial processes (cf. (Ćalić, 2011).

An opportunity to shed new light on such relationships has arisen in an evaporite karst setting at the margins of the shrinking Dead Sea (Yechieli et al., 2016). There, clusters of tens to over a hundred sinkholes (1-75 m diameter) that are surrounded by larger scale (100-800 m diameter) depressions have rapidly developed over the last 40 years (Al-Halbouni et al., 2017; Atzori et al., 2015; Filin et al., 2011). In particular, recent studies by (Al-Halbouni et al., 2017; Watson et al., 2018) involving field work, remote sensing and photogrammetric surveying enabled the detailed documentation of spatio-temporal relationships between sinkhole and depression development at Ghor Al-Haditha, on the eastern shore of the Dead Sea (Figure 1A). The area exhibits mature karst landforms comprising individual and compound sinkholes. The term compound sinkhole here means the nested or non-nested coalescence of individual sinkholes. Sinkhole clusters or aggregations, commonly comprise multiple individual sinkholes and/or compound sinkholes in close proximity. Such clusters commonly lie within gentler, larger-scale (uvala-like) depressions of up to several hundred of meters in diameter, as depicted in Figure 1B and C.

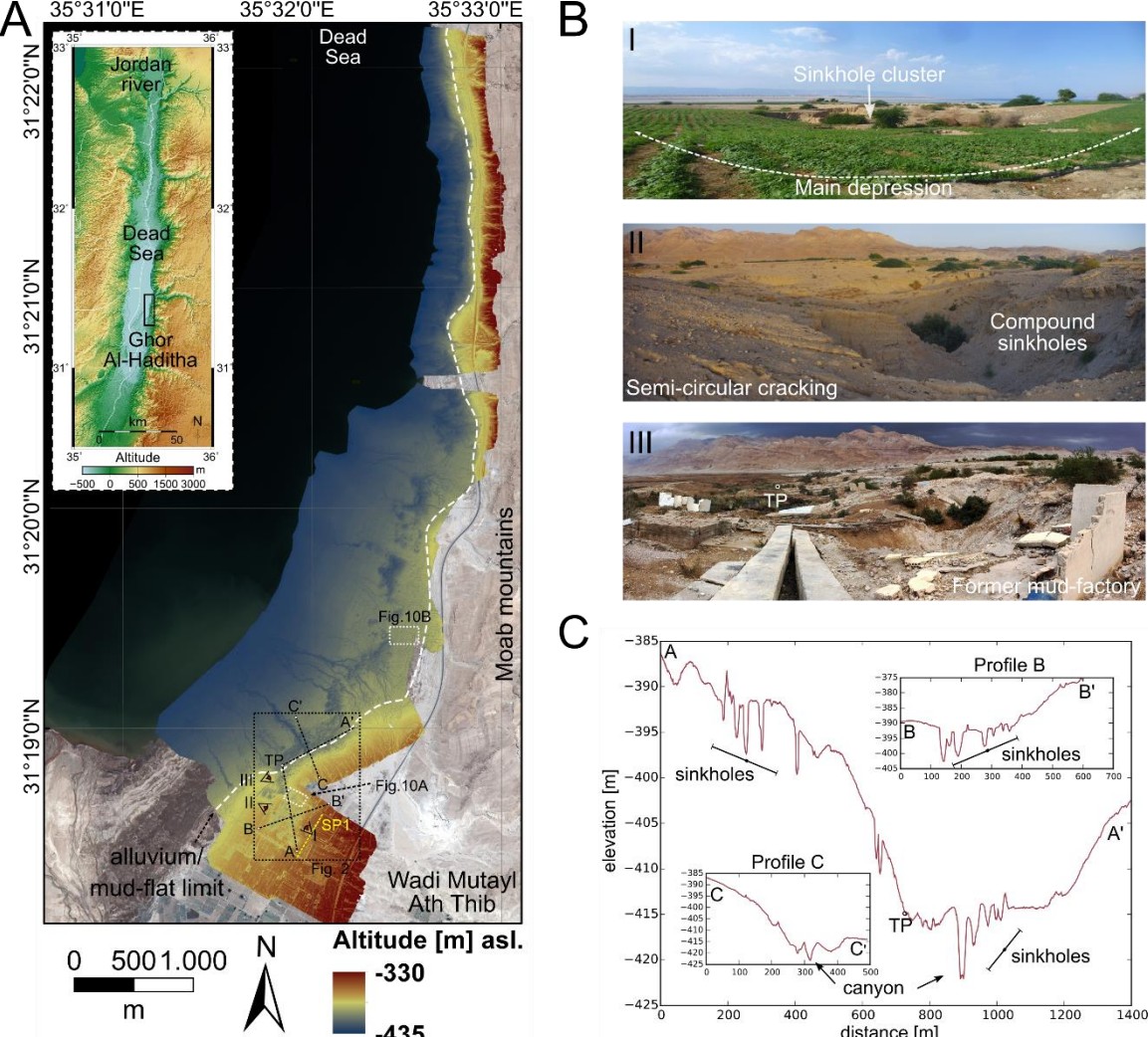

**Figure 1: Topography of the Ghor Al-Haditha sinkhole are. (A)** Digital surface model (DSM) from 2016 on Pleiades satellite image from 2015 for the sinkhole area at the Dead Sea (inset). The DSM has a resolution of 10 cm px$^{-1}$ and accuracy of 37 and 31 cm (H and V), respectively. The main zone affected by sinkholes extends from S towards the NNE along the contact zone between alluvial fans and the mudflat (dashed white line) and comprises of several partly connected large depression zones, of which the main area is indicated by the square. SP1 is the a section of seismic profile one of (Polom et al., 2018). **(B)** Typical examples of sinkhole formations in the main depression. **(I)** View from the stable agricultural area towards the centre of the depression. **(II)** Nested sinkholes and ground cracks from the opposite view. **(III)** Destroyed "Numeira" mud factory at the turning point (TP) of the depression. **(C)** N-S and E-W topographic profiles across the several hundred meter depression, derived from the DSM of 2016.

Initially, these karst landforms develop as small localised subsidence zones, with single sinkholes that form in heterogeneous material made of Dead Sea mud, alluvial fan sediments and salt ((Watson et al., 2018). Wider-scale subsidence and sinkhole clustering follows, with ground fracture systems developing that are geometrically related to the larger-scale depression rather than to the individual sinkholes or sinkhole clusters (Figure 2).

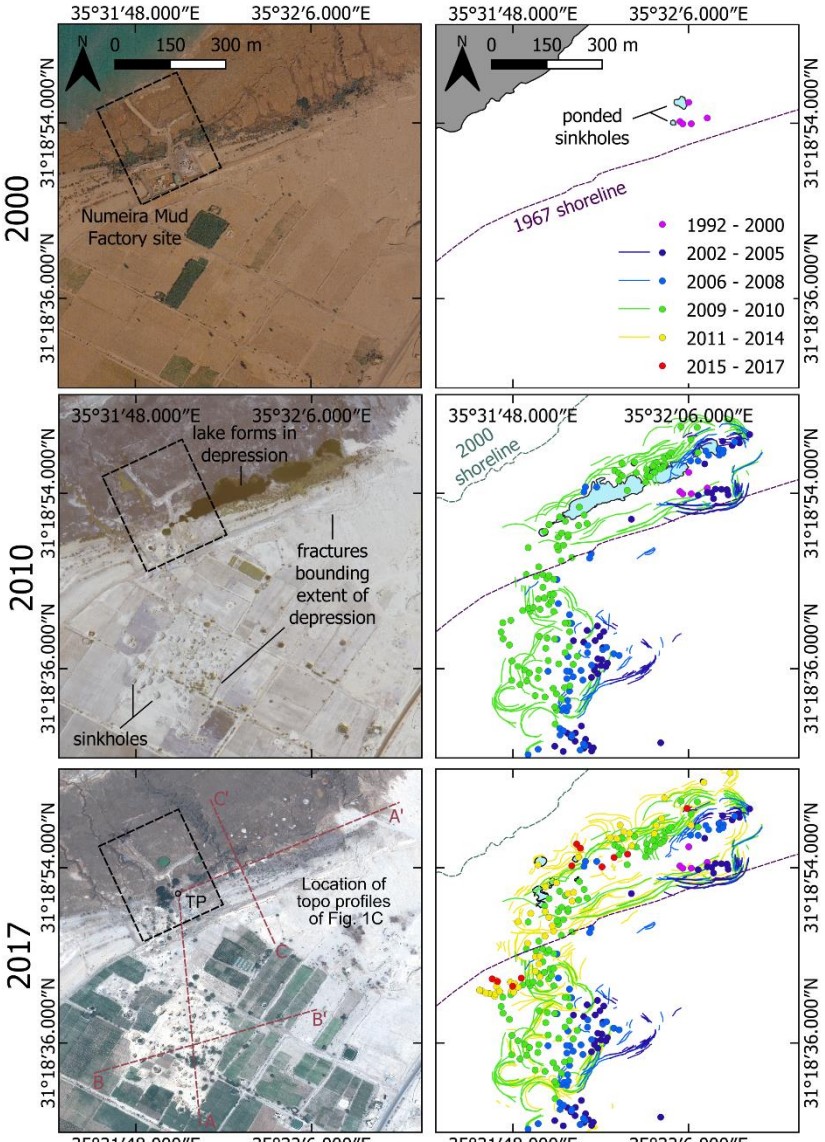

**Figure 2: Remote sensing analysis of the evolution of sinkholes, cracks and large depressions at the main depression of Ghor Al-Haditha, Dead Sea. Small single sinkholes appear in 2000 at the former "Numeira" mud factory site (0.6 m/pix aerial photo from Royal Jordanian Geographic Centre). Up to 2010, lakes, sinkhole clusters and large fractures have appeared around a depression zone spanning over both alluvium and mud-flat (0.5 m/pix GeoEye-1 satellite image). Up to today, the number of fractures and sinkholes as well as the depth of the depression has increased (0.5 m/pix Pleiades satellite image). Red lines in lower left image mark the locations of the profiles shown in Figure 1C.**

In this paper, we test the hypothesis that the driving process for the geomorphological and structural development of such large-scale karst features is a widespread, differential subsidence above an array of sub-surface cavities, with temporally- and spatially variable patterns of material removal driven by base-level fall associated with the shrinking of the Dead Sea.

To test our hypothesis, we use novel 2D distinct element method (DEM) numerical modelling. We examine two end-member growth scenarios of model cavity arrays, and we look at the surface morphologies, sub-surface structure and stress patterns developed by subsidence of the overburden as those cavity arrays grow. The numerical results are discussed with respect to both surface and subsurface data from the Dead Sea evaporite karst. Interpretation of shear wave reflection data indicates that the subsurface under the alluvial fan sediments at the Ghor Al-Haditha site is characterised by inclined layering typical of a pro-grading Gilbert-type delta, superimposed on which are zones of disrupted seismic reflectors, as well as bowls and depression structures (Polom et al., 2018). We provide in this work both a qualitative and quantitative comparison of the results from the seismic survey and from the numerical modelling. We show that our more complex end-member modelling scenario is able to explain complementary observations from surface morphology to subsurface hydrology and subsurface geophysics.

## 2    Numerical approach

### 2.1    Distinct Element Method Numerical Modeling

The mechanical interaction of a single void space with its surrounding rock mass has been investigated previously by analytical methods (e.g. Tharp, 1999) and numerical modelling studies (e.g. Al-Halbouni et al., 2018; Baryakh et al., 2009; Fazio et al., 2017; Hatzor et al., 2010; Parise and Lollino, 2011). Little is known, however, about the mechanical interactions between multiple, actively-evolving void spaces in the subsurface and about how these interactions may lead to the development of sinkhole clusters and large-scale depressions. Moreover, commonly-used continuum numerical simulation methods are usually not appropriate to simulate rotation, detachment and non-continuous deformation found in rocks or semi-consolidated materials that have been subject to large strains, such as are characteristic of sinkhole collapses. Discontinuous medium simulation methods, on the other hand, allow for complex behaviours like spontaneous crack formation and block rotation (Jing and Stephansson, 2007). Distinct Elements (Cundall, 1971) is a subset of Discrete Element Modeling (Cundall and Strack, 1979; Jing and Stephansson, 2007), whereby a material is represented as an assembly of non-deformable particles in the shape of disks of unit thickness (2D, Figure 3) or spheres (3D). The particles are assigned a density, radius and an elastic contact modulus. They are assembled with a certain porosity and follow a defined size distribution. The particles follow the Newton-Euler laws of motion and the linear force-displacement law as they interact elastically at each contact point. The assembly is generated via a randomised particle packing scheme and a gravitational settling scheme (Al-Halbouni et al., 2018), after which particles can be bonded with their neighbours (Potyondy and Cundall, 2004). In this study, we used the parallel-bond model (PBM) in the commercially available software PFC2D (Potyondy, 2014), which sets a second pair of elastic springs that incorporate moments and can fail either in shear or tension, allowing for a complex elasto-plastic rheology (Al-Halbouni et al., 2018; Holohan et al., 2011; Potyondy and Cundall, 2004; Schöpfer et al., 2009). The resulting differential equations are solved via a finite-difference, explicit, time-stepping algorithm (Jing and Stephansson, 2007). Each model requires multiple realisations as the outcomes generally depend on the randomised particle packing. For more mathematical

details on the calculations and modelling scheme refer to (Al-Halbouni et al., 2018; Itasca, 2014; Jing and Stephansson, 2007; Potyondy, 2014; Wang et al., 2018).

## 2.2    Cavity growth in a DEM model

Al-Halbouni et al. (2018), simulated the growth of a single cavity in a DEM model, and they conducted a detailed calibration and verification procedure to determine the optimal model geometry, resolution and material removal technique. We here adopt the same setup parameters and conditions: i.e. a 2D box of model height H x width W = 400 x 400 m, a uniform particle distribution between a minimum ($R_{min} = 0.24$ m) and maximum particle radius ($R_{max} = 0.4$ m); an initial porosity of the unsettled assembly of n = 0.2; no-slip boundary conditions; and a fixed wall elastic modulus of $E_w = 5$ GPa. Instead of simulating material removal in a single cavity as in Al-Halbouni et al. (2018), however, we here implemented an array of multiple cavities of arbitrary shape (Figure 3, cf. Appendix A.1). The adopted procedure is an incremental particle removal that mimics subrosion, i.e. the removal of subsurface material by chemical leaching and/or physical erosion. Our 2D model thus represents a flow-perpendicular cross-section through a groundwater conduit network, which we envisage to result from dissolution that rapidly localises through a feed-back mechanism of enhanced fluid flow with increasing dissolution (Weisbrod et al., 2012) and which in turn can also promote conduit growth by physical erosion.

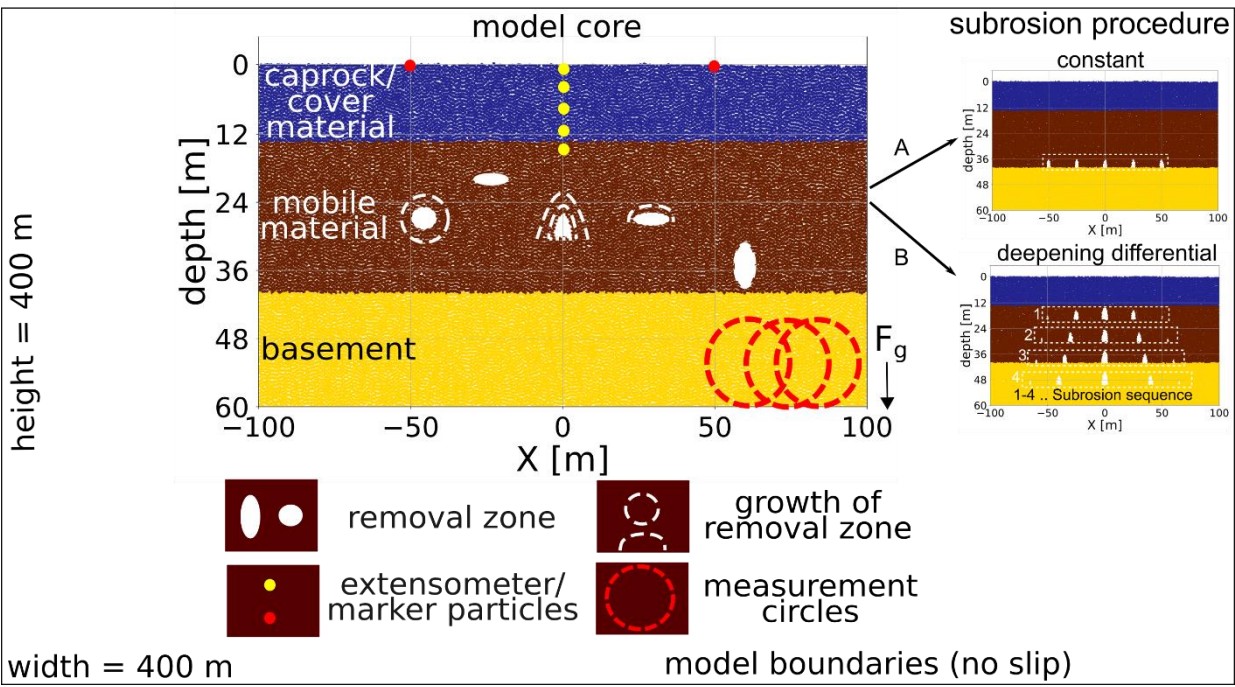

**Figure 3: Generic setup for multiple cavity modelling with DEM. The core of the model and the specific setup of the void zone and implemented features are shown here. Arbitrary material removal zones can be defined and associated with different removal functions activated at arbitrary removal zone growth increments. The subrosion procedures for (A) models with deep constant subrosion and (B) models with deepening and differential cavity growth are highlighted.**

We test two end-member growth scenarios of the cavity array and their impact on surface subsidence in the models: (1) cavity growth at the same depth level and at the same individual growth rate (Figure 3A); (2) cavity growth at progressively deepening levels with varying individual growth rates (Figure 3B). The quasi-static growth is simulated by incremental particle removal and details can be found in Appendix A.1A.2. In the first scenario, five semi-elliptical cavities begin to grow at the same time, at the same constant rate, and at the same depth of 40 m. The latter has been chosen according to tests on single cavities in Al-Halbouni et al. (2018), and similar tests on cavity arrays as presented in Appendix A.2. In the second scenario, the five cavities start to grow simultaneously, but the initial cavity area is largest in the centre and decreases laterally. In addition, the cavity growth rate is largest for the central cavity and smallest for the outermost cavities. This represents the energy distribution of a progressively focussed flow within the growing conduit system. Furthermore, the array geometry changes as new cavity sets develop at progressively increasing depths from 20 m to 50 m at 10 m increments. The growth in the shallower cavity set stops when the new set initiates. The area of removed particles multiplied by a unit thickness is considered as the total removed volume $\Delta V$. A 30 m deep cavity set only initiates after a total volume removal of $\Delta V \sim 400$ m³, a 40 m deep set starts after $\Delta V \sim 800$ m³ and a 50 m deep set starts after $\Delta V \sim 1200$ m³. The width of the array also increases slightly from ~110 m in the shallow part to ~150 m in the deep part. This progressive initiation of newer and deeper sets of cavities represents a vertical evolution of a dissolution front during base-level fall, the main hydrogeological boundary condition at the shrinking Dead Sea (Abelson et al., 2017; Bartov, 2002; Watson et al., 2018).

## 2.3    Material parameters

The bonded particle assembly's bulk material properties, which emerge from the properties defined on the particle scale, were constrained by simulated geomechanical tests on material samples (Schöpfer et al., 2007; Al-Halbouni et al., 2018). Parallel-bond tensile strength, modulus and friction, cohesion and friction angle, as well as contact modulus and friction, are hence transferred to corresponding bulk values of unconfined compressive strength (UCS) and tensile strength (T), Poisson's ratio (ν) and Young's modulus (E). This calibration procedure has been done for three materials representing those in which sinkholes form at the Dead Sea shoreline: (1) low-strength marl (mud) of the former Dead Sea lakebed, (2) middle-strength sandy-gravel alluvial fan sediments and (3) relatively high-strength Holocene rock salt of the Dead Sea (Table 1). For (1) a bond healing procedure has been implemented to account for a more realistic recombination behaviour of naturally wet muddy material. For each material, the calibration was run on ten subsamples of H x W = 10 x 8 m size, with approximately 200 particles of a mean radius $\bar{R} = 0.32$ m. See Al-Halbouni et al. (2018), for details on the procedure.

Table 1: Estimated mean bulk geomechanical properties of the main materials in sinkhole-affected areas at the Dead Sea. The variation of the bulk strength values is related to analysis by both Mohr-Coulomb and Hoek-Brown failure criteria assuming intact rock (Al-Halbouni et al., 2018). Note the geotechnical engineering convention of compressive stress being taken as negative.

| Parameter | Symbol | Unit | Wet lacustrine mud | Alluvial sediment | Holocene Salt |
|---|---|---|---|---|---|
| Particle packing porosity | $n_{eff}$ | - | 0.21 | 0.2 | 0.17 |

| Bulk density | $\rho_{bulk}$ | [kg/m$^3$] | 2145 | 2200 | 2075 |
|---|---|---|---|---|---|
| Young's modulus | $E_{eff}$ | [GPa] | $0.084 \pm 0.02$ | $0.174 \pm 0.025$ | $1.106 \pm 0.126$ |
| Poisson's ratio | $\nu_{eff}$ | - | $0.19 \pm 0.12$ | $0.31 \pm 0.6$ | $0.30 \pm 0.03$ |
| Unconfined compressive strength | UCS | [MPa] | -0.25 to -0.06 | -0.92 to -0.52 | -1.54 to -1.23 |
| Unconfined tensile strength | T | [MPa] | 0.01-0.2 | 0.18-0.24 | 0.31-0.43 |
| Cohesion | c | [MPa] | 0.11 | 0.18 | 0.36 |
| Friction angle | $\phi$ | [°] | 5.7 | 22.3 | 28.8 |

## 2.4  Geophysical parameter tracking

Distributed measurement circles of 10 m diameter (area $A^m = 78.5$ m$^2$) are used to record stresses, strain rates, positions and porosity of the particle assemblies (see Al-Halbouni et al. (2018), for details). From recording of the stress components $\sigma_{xx}, \sigma_{xy}, \sigma_{yx}, \sigma_{yy}$ and particle areas, we calculate the principal stresses $\sigma_1$ (most compressive, i.e. most negative) and $\sigma_3$(least compressive) The maximum shear stress is calculated via half the differential stress (e.g. Holohan et al., 2015):

$$\tau_{max} = \frac{(\sigma_3 - \sigma_1)}{2} \tag{1}$$

For strain calculation, the displacement gradient tensor is calculated for particles inside the fifty percent overlapping measurement circles between two simulation stages (e.g. Schöpfer et al., 2006). The maximum in-plane shear strain is determined via the principal strains $\varepsilon_1$, $\varepsilon_3$ and the shear strain $\gamma_{xy}$:

$$\gamma_{max} = 2\sqrt{\frac{(\varepsilon_1 - \varepsilon_3)^2}{2} + \frac{\gamma_{xy}^2}{2}} \tag{2}$$

We use porosity tracking results to determine apparent elastic moduli, which can then be translated via bulk density into apparent bulk seismic velocities. In general, for a homogeneous, linearly-elastic, isotropic medium, compression wave velocities ($v_p$) and shear wave velocities ($v_s$) are estimated by:

$$v_s = \sqrt{\frac{G}{\rho}} \tag{3}$$

$$v_p = \sqrt{\frac{K + \frac{4}{3}\nu}{\rho}} = \sqrt{\frac{2G(1-\nu)}{\rho\,(1-2\nu)}} \tag{4}$$

$K, G$ are the bulk/shear modulus, respectively. $E = 2G(1 + \nu)$ is Young's elastic modulus for homogeneous, isotropic materials, with $\nu$ as the Poisson ratio. $\rho$ is the bulk density calculated by $\rho = \rho_{particle}(1 - n)$, with n as the particle packing porosity. A correction factor is needed to account for the differences between static and dynamic moduli to enable a comparison of numerical simulation with field data. Dynamic field methods like seismic reflection profiling measure at

small strains and therefore reveal high values of the shear modulus. E and ν of the model materials are known from simulated large-strain compression tests for a variation of confining pressures and porosities (Al-Halbouni et al., 2018). We here use $G_{dyn} \sim 1.5 * G_{stat}$ , the dynamic shear modulus, approximated as a minimum scaling of the static shear modulus determined for unconsolidated sand in a cycling loading/unloading and shearing test (Soldal and Mondol, 2015). The factor depends on the applied static technique in laboratory experiments and on the cycles, the difference rises mainly from the strain amplitude (Hammam and Eliwa, 2013; Wichtmann and Triantafyllidis, 2009). Furthermore, from the simulated compression tests, conservative values for moduli and Poisson ratio are taken at limits where few or no cracks have appeared in the sample. For a more realistic approach, the values are further adjusted by accounting for the crack (broken bond) density in the model, following *Dahm and Becker*, (1998). For the adjustment to the DEM, crack density is defined as $c = k\pi\bar{R}/A^m$, with $k$ as the number of cracks, $A^m$ the area of the measurement circle and $\bar{R}$ as the mean particle radius, which is a proxy to the parallel-bond (crack) half length (see Al-Halbouni et al. (2018), for details). Cracks, i.e. broken parallel bonds in DEM, are recorded by using an intrinsic "fishcall" procedure (Hazzard, 2014; Hazzard and Young, 2004), and distributed onto the according measurement circles. With increasing crack density, $\nu$ and the apparent (effective) shear modulus $G_{eff}$ are expected to change by:

$$\nu(c) = \frac{(1-\nu_0)e^{\frac{fc}{2}}+2\nu_0-1}{2(1-\nu_0)e^{\frac{fc}{2}}+2\nu_0-1} \tag{5}$$

$$G(c)_{eff} = G_0/(2(1-\nu_0)e^{\frac{fc}{2}}+2\nu_0-1) \tag{6}$$

For randomly oriented cracks the mean of the shear tractions on the cracks is one half of the maximum shear stress in the body (cf. Dahm and Becker, 1998), for which a factor of f = 0.5 can be estimated. Furthermore, $\nu_0 = 0.5$ is the Poisson ratio when the bulk modulus is larger than $G_0$, the shear modulus of a homogeneous, isotropic rock mass (Dahm and Becker, 1998).

## 3    Modeling Results

In this section we present outcomes of both end-member cavity growth scenarios, while focussing on a final model setup that most closely reproduces the natural karst landforms (Sec. 1). For both end-member cavity growth scenarios, we also show the results of models for layered combinations of weak and strong materials common at the Dead Sea shoreline. Note that for certain model conditions (weak material and/or deep subrosion - see also Al-Halbouni et al., 2018) the cavity walls and overburden tend to collapse immediately, and so cavities may remain small during a model evolution or may exist only instantaneously for each increment of material removal.

In Figure 4 we compare the outcomes of both end-member cavity growth scenarios for four different material setups representing weak and strong overburden configurations: (I) alluvium on lacustrine mud (Figure 4A), (II) a thin salt-layer above lacustrine mud and alluvium (Figure 4B), (III) pure lacustrine mud (Figure 4C) and (IV) a mud layer above a salt and alluvium succession (Figure 4D). In this overview, a clear difference between cavity growth scenario (1), constant medium-

depth (40 m) subrosion, and scenario (2), a differential deepening subrosion, can be seen. While scenario (1) results in block-wise subsidence or large-scale sagging over the entire array, scenario (2) reproduces the observed pattern of multiple sinkholes in a large-scale depression zone. The main structural and morphological features that relate to differences in material and in subrosion scenario are marked in each individual plot.

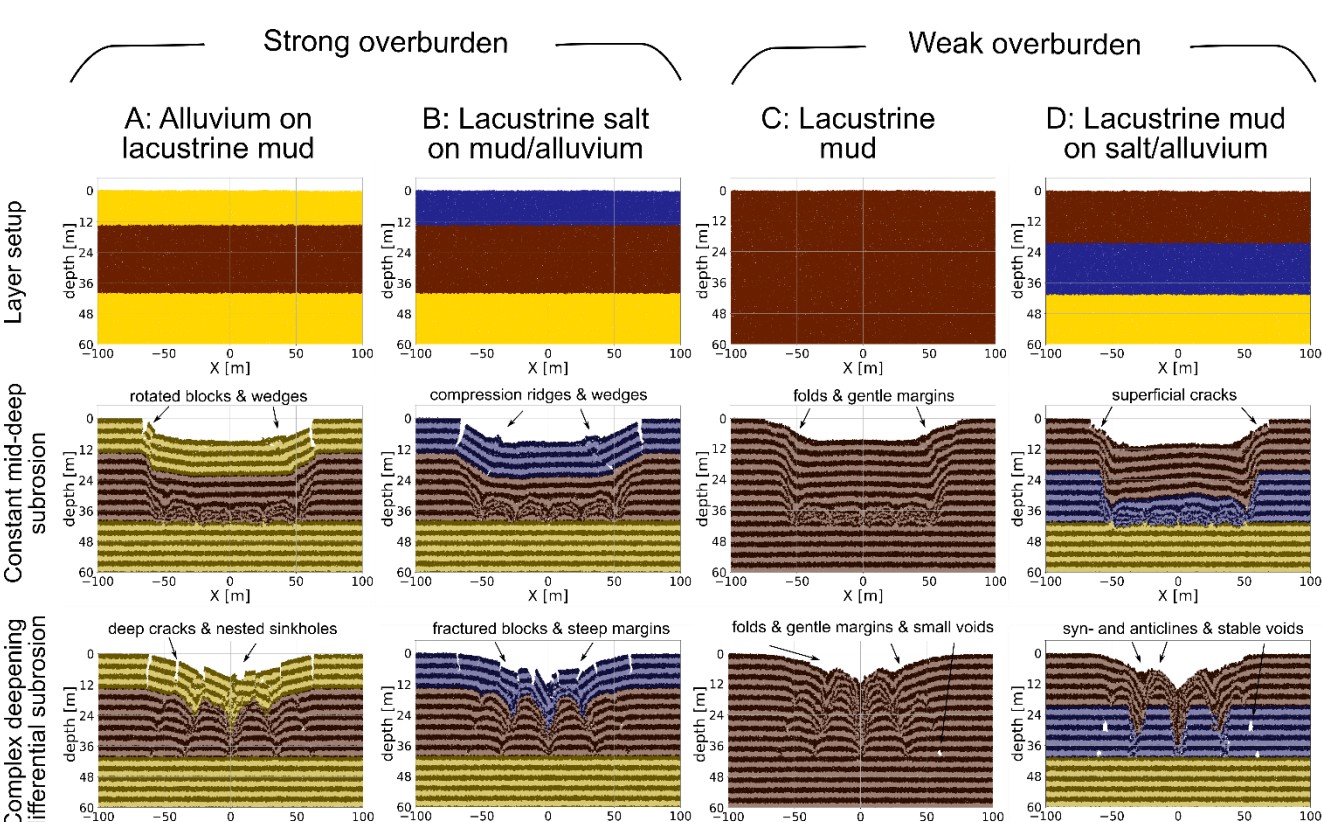

**Figure 4: Comparison between final results for two cavity growth endmembers and different material compositions common at the Dead Sea shoreline. The removed volume at the shown stages is approximately 900 m³. Strong overburden: (A) alluvium/mud succession and (B) salt on mud/alluvium succession. Weak overburden: (C) pure lacustrine mud and (D) mud on salt/alluvium succession. Note that passive marker layers are applied to highlight structural features.**

In Figure 5 we show the main evolutionary stages I-VI of sinkhole/depression development for cavity growth scenario (2), i.e. the deepening differential cavity growth scenario. Detailed animations of the evolution can be found in the electronic supplement.

15    For all combinations of material type, the large-scale depression is deepest throughout the evolution above the central and fastest growing cavity in each array (as per definition in the model setup).

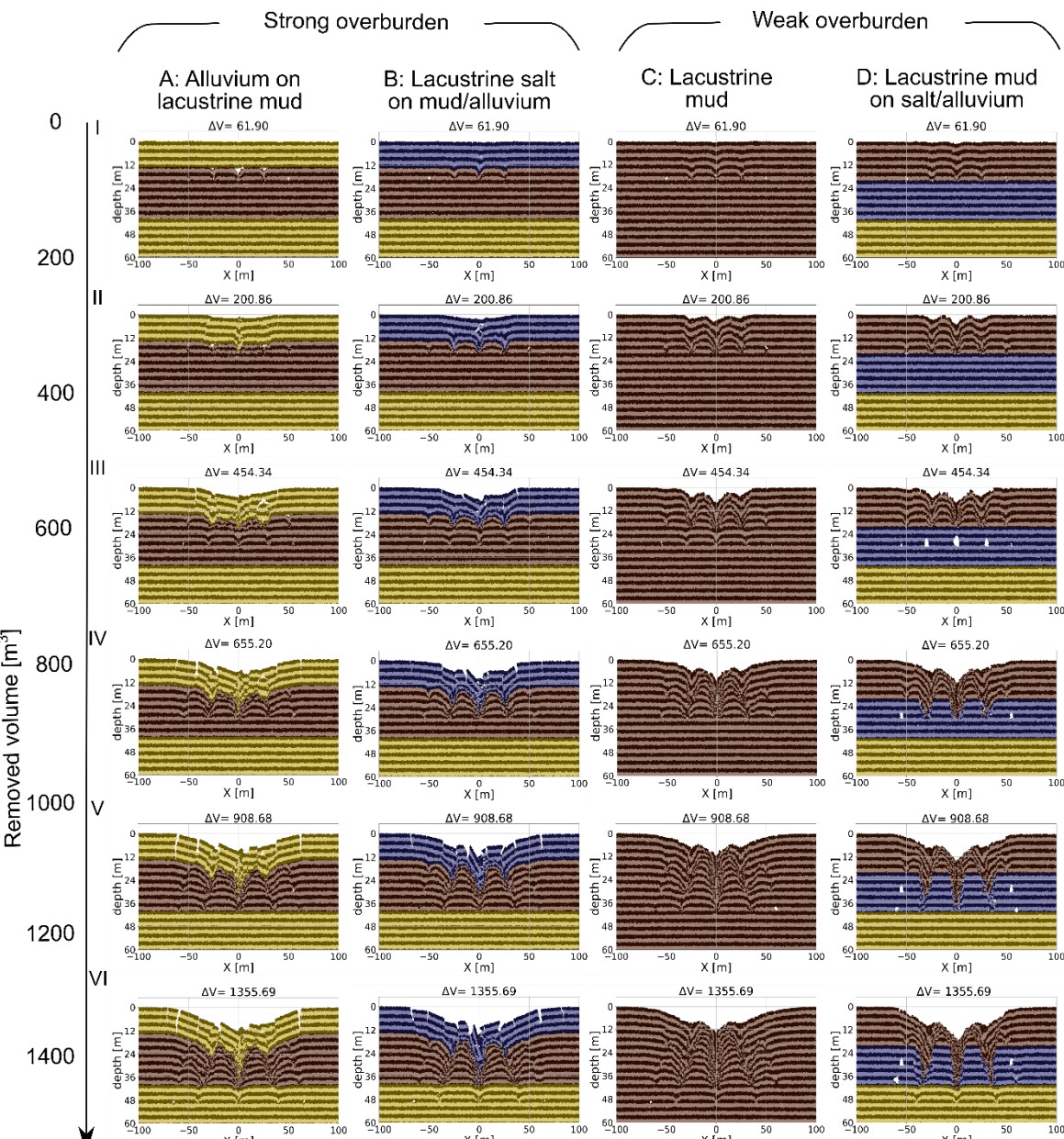

**Figure 5: Final model sinkhole evolution results for four different material combinations common at the Dead Sea shoreline. Strong overburden: (A) alluvium/mud succession and (B) salt on mud/alluvium succession. Weak overburden: (C) pure lacustrine mud and (D) mud on salt/alluvium succession. The removed volume [m³] is shown above the plots. Note that passive marker layers are applied to highlight structural features.**

In general, for the material combination of a strong or weak overburden above a weak cavity-hosting material (Figure 5A-C), individual sinkholes form synchronously with, or just before, the development of a larger-scale synclinal depression that initially spans several central cavities and eventually spans the cavity array as a whole. The formation of the sinkholes more

clearly predates the array-scale depression where the overburden is weak. The margins of the array-scale synclinal depression are commonly delimited, especially in the strong overburden, by fractures and/or faults. These marginal fractures geometrically relate to subsidence across several cavities, or to subsidence across the entire array, rather than to collapses into individual cavities. In weaker overburden, the margins of the main depression are defined by inward bending (sagging) of the overburden layers (although in detail there are many small-scale fractures here).

For the material combination of a weak overburden above a strong cavity-hosting material (Figure 5D), large cavities can develop before the overburden collapses into them. This produces deeper and wider sinkholes in the later stages of the model evolution. Also in this case, the strong cavity-hosting material does not deform so easily around the cavity array as a whole; therefore, synclinal bending of the overburden across the cavity array is much less pronounced. Consequently, a larger-scale depression forms in this case mainly by nesting and coalescence of the sinkholes.

For individual sinkholes in strong overburden materials (Figure 5A, B), the collapsed overburden is commonly delimited by faults near the surface, but at depth the structure takes a synclinal form (V-shaped) on the same scale as the individual cavities. For individual sinkholes in weak overburden material (Figure 5C, D), the collapsed overburden shapes are synclinal at all depths. In the strong cavity-hosting material (Figure 5D), the deep-levels of the individual collapse zones are again in part fault-bounded but also take in part a synclinal form. These cavity-scale synclinal structures represent the downward flow of the weak material into the cavities or, where cavity formation is inhibited, into the zones of material removal.

Depth to diameter (De/Di) ratios of simulated sinkholes and depressions are given in Table 2 as mean values of five model assemblies of each material combination. The dimensions of depressions at the scale of the entire cavity-array range from ~ 65-190 m across and ~ 2 – 18 m deep, while individual sinkholes have dimensions of ~ 1.5-36 m across and ~ 0.5-12 m deep. Higher De/Di ratios of 0.48 – 0.64 for sinkholes are generally recorded for cover material of higher strength (alluvium, salt), while lower De/Di ratios of 0.22-0.24 are found for low strength cover material (mud). The De/Di ratios of 0.08-0.14 of the larger-scale depressions are many times lower (in some cases nearly an order of magnitude lower) than those of the sinkholes.

The evolution of depth and diameter of large-scale depressions (Figure 6) shows the influence of the material strength on their geometries. A clear divergence can be observed between mud subsurface and salt subsurface models. A mechanically weak subsurface (mud) enables a lateral widening of the depression at the expense of deepening. A mechanically strong (salt) subsurface inhibits the synclinal bending at the margins of the main depression, leading to deepening of the depressions and preventing their widening.

**Table 2: Depth to diameter ratios of simulated array-scale depressions. The average results for the four different material setups and different stages of depression development are given. The depth of a depression is hereby considered the deepest point, which might coincide with the deepest point of a sinkhole within the depression. The diameter goes as far as a vertical surface displacement of ~10 cm amplitude can be observed. 5 realisations were done for each material combination.**

| Type/Model setup | Lacustrine mud | Alluvium on mud | Salt on mud/alluvium | Mud on salt/alluvium |
|---|---|---|---|---|
| Depression - early stage | $0.05 \pm 0.003$ | $0.03 \pm 0.009$ | $0.04 \pm 0.003$ | $0.07 \pm 0.006$ |
| Depression - middle stage | $0.08 \pm 0.004$ | $0.07 \pm 0.013$ | $0.08 \pm 0.006$ | $0.12 \pm 0.010$ |
| Depression - late stage | $0.08 \pm 0.004$ | $0.09 \pm 0.004$ | $0.11 \pm 0.007$ | $0.14 \pm 0.005$ |
| | | | | |
| *Sinkholes - final stage* | $0.22 \pm 0.12$ | $0.48 \pm 0.36$ | $0.64 \pm 0.3$ | $0.24 \pm 0.08$ |

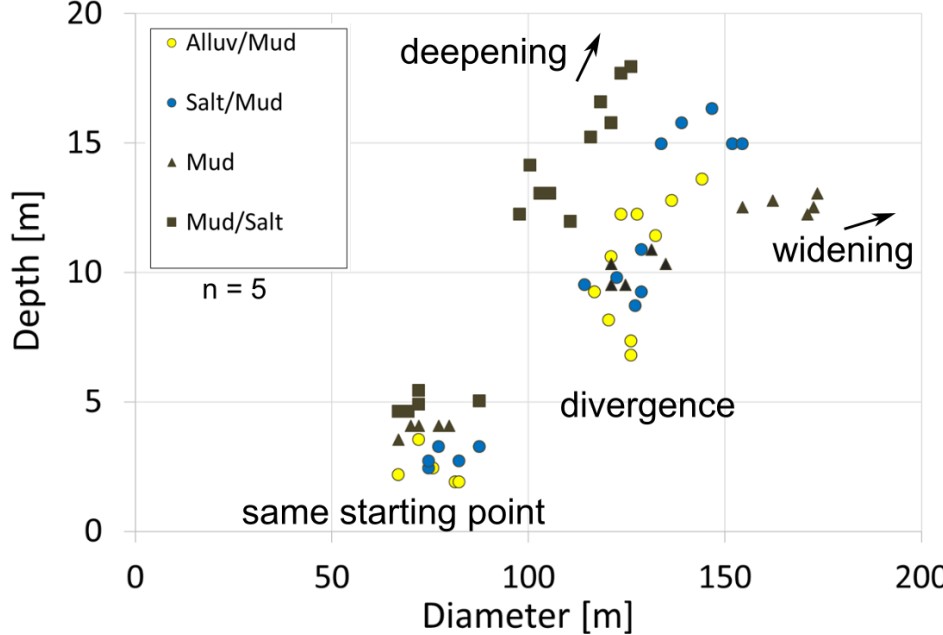

**Figure 6: Depth versus diameter for different stages of the final model large-scale depressions for different material combinations. A clear divergence can be observed between mud-rich subsurface and salt-rich subsurface models at the late stage of the simulation. The number of realisation of each model is $n = 5$.**

10      The influences of different positions and different speeds of material removal zones have also been tested thoroughly (see Appendix A for details). In all material cases for scenario (1), i.e. the constant cavity growth level and rate, and regardless of the depth of the array, only large, array-scale depressions occur and no sinkholes form in relation to the individual cavities. A shallower cavity array leads only to faulting/segmentation of the sinking block and/or fracturing of the margins. Varying the speed of array-wide subrosion produces no discernible difference in model outcome, as expected for the quasi-static approach.

Clearly, a differential cavity growth is essential for development of sinkholes within a larger-scale depression. This is even more pronounced with accelerating growth of the central cavities. Additionally, and importantly, for reproducing the morphological features and the order of appearance of sinkholes relative to the larger-scale depression, as observed in the Dead Sea examples, a simulated deepening of the karstification/subrosion level, i.e. cavity growth scenario (2), is necessary. From comparison of numerical simulations of all tested scenarios and setups in the previous section and in Appendix A.1 and A.2, we conclude also that the inter-cavity distance has an influence on the sinkhole clustering and generation of larger-scale depressions. In the limit, if the inter-cavity distance is wide enough, no clusters or large-scale depressions would form but only individual sinkholes.

### 3.1    Stresses and strains in a multiple void space system

The differences in model outcome depending on the cavity growth scenario are better understood when looking at the stress and strain distribution patterns. The maximum shear stress $\tau_{max}$ around the cavity arrays in model cavity growth scenarios (1) and (2) is shown in Figure 7. We here compare two different material setups, strong alluvium on weak mud and weak mud on strong salt and alluvium. Each model has the same particle assembly and comprises five void spaces at ~ 40 m depth at stages immediately before or exactly during the collapse of the overburden. The differential subrosion scheme here uses the same setup as in the model Figure 14G in Appendix A.2, which is without a deepening of the subrosion zone, to avoid effects of remnant stress distributions.

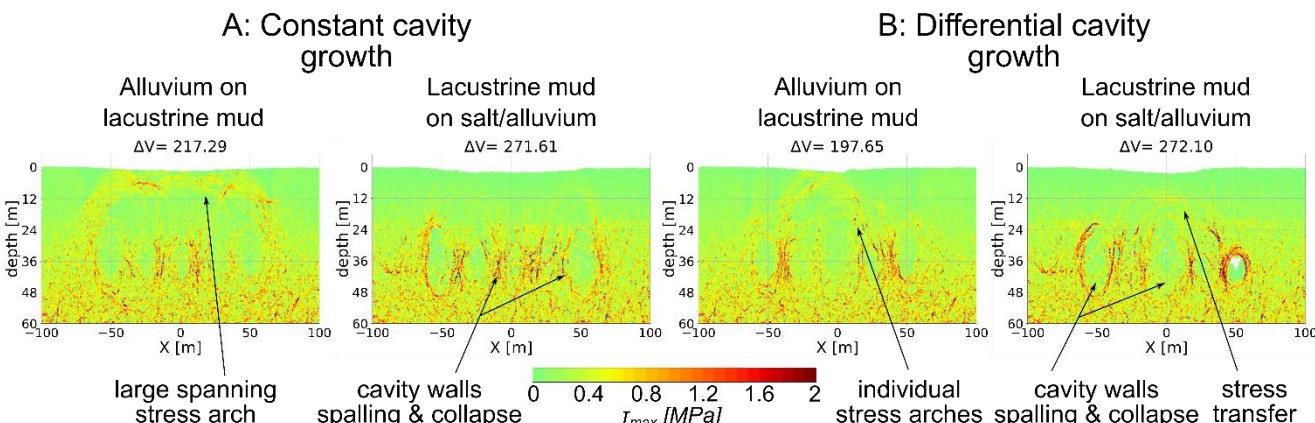

**Figure 7: Maximum shear stress around void spaces for (A) constant and (B) differential cavity growth scenarios models. Chosen are two material combinations where the subrosion affected layer differs in strength: alluvium on mud multilayer and mud on salt/alluvium succession. Shown are critical stages after void space installation followed by or exactly during overburden collapse for the same particle assembly. The removed volume [m³] is shown above the plots.**

Regarding the mechanical development, cavity growth scenario (1) produces a stress arch spanning the whole array of cavities, best visible in the alluvium on mud combination of Figure 7A. For cavity growth scenario (2) produces a more complex pattern of more localised stress concentrations and arches appearing around or between individual cavities. The setup of a constant

cavity growth rate hence leads to a block-wise subsidence, while for differential cavity growth rate, the interaction of stresses around and between the cavities leads to multiple sinkhole development in a large-scale depression. Appendix A.3 shows similar results for principal stresses.

For the same scenarios, the maximum shear strain $\gamma_{max}$ is shown in Figure 8. It highlights the different subsidence styles: block-wise subsidence for a constant subrosion scenario (1), and fragmented, individual overburden failure with faulting/segmentation for the differential subrosion scenario (2), with stable areas of low strain in between the subsiding blocks. In comparison to the shear stress images, this representation more clearly illustrates cracking and fracture development. These observations are complemented by the incremental strain and maximum shear stress evolution as presented in the Appendices A.4 and A.5.

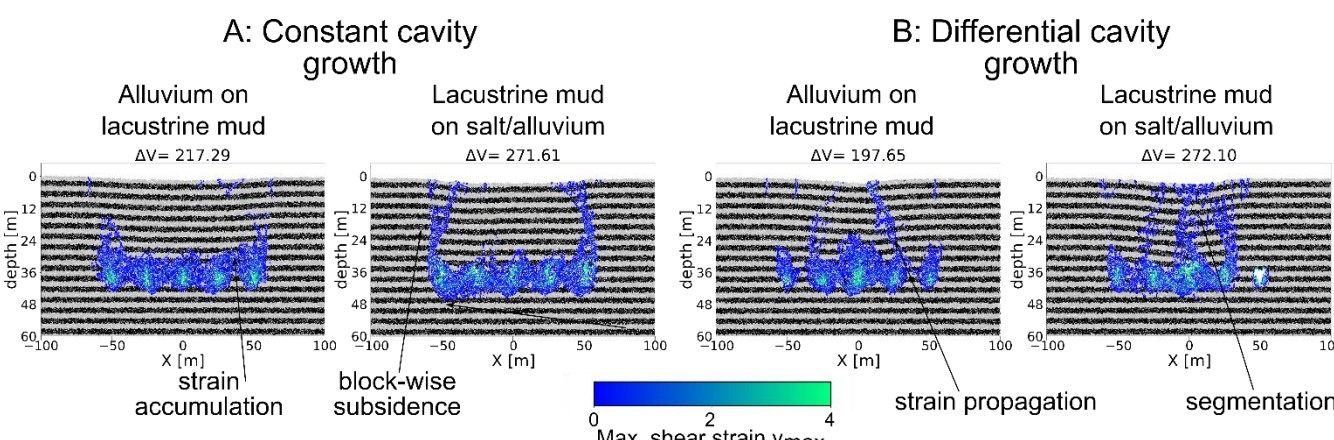

**Figure 8: Maximum finite shear strain for (A) constant and (B) differential cavity growth scenarios models, Chosen are two material combinations where the subrosion-affected layer differs in strength: alluvium on mud multilayer and mud on salt/alluvium succession. Shown are the same critical stages after void space installation as in Figure 7. The removed volume [m³] is shown above the plots.**

## 3.2 Generic geophysical parameters

Figure 9 shows the synthetic geophysical parameters that characterize the model underground. As we consider a non-elastically deformed underground, all derived elastic parameters must be regarded as apparent. We concentrate on a snapshot of the final stage of the alluvium-on-mud model simulating cavity growth scenario (2) – see Figure 5A at a removed volume of $\Delta V = 1355\ m^3$ for the most important structural features. A deep and large depression zone with sinkholes has formed already. At this stage, the actively growing cavity set (or active subrosion zone) lies at ~ 50 m depth. The initial porosities lie between 0.2 and 0.1 depending on the depth (Figure 9A). Areas of porosities over 0.5 we consider as "empty" space with a zero modulus/seismic velocity. Initial values of the shear wave velocity in "stable" ground are 100-150 m/s for mud and 200-450 m/s for alluvium (Figure 9B).

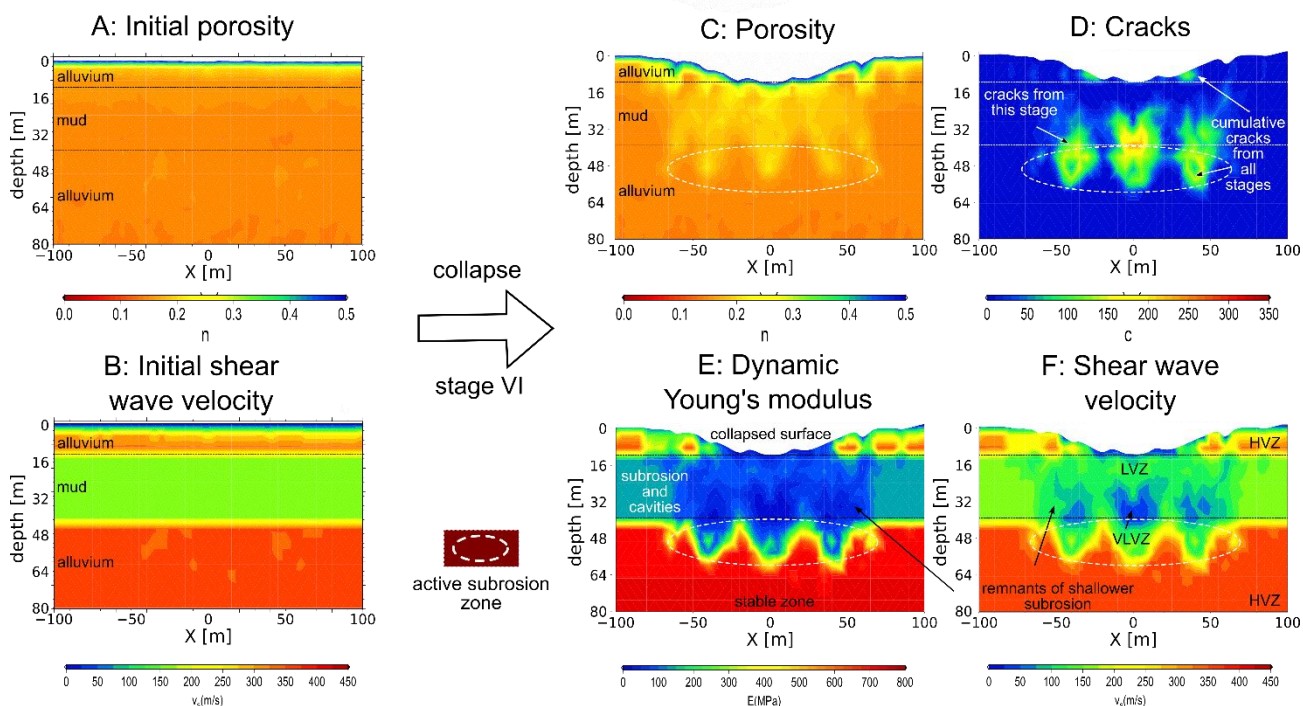

**Figure 9: Geophysical characterization of the underground. Derived generic parameters for initial and final model alluvium on mud shortly before stage VI (cf. Figure 5A). (A) Porosity & (C) distribution. . (B) & (F) Apparent shear wave velocities. (D) Number of cracks in the vicinity of the subrosion zone. Note that the cracks in the mud layer arises from the current stage of material removal (due to rebonding), whereas those in alluvium layers are accumulated throughout the model evolution. (E) Apparent dynamic elastic (Young's) modulus. Black lines mark the initial limits of the mud horizon. The surface has collapsed partly into a depression plus sinkholes.**

The porosity distribution at the final stage can be seen in Figure 9C. The number of cracks is depicted in Figure 9D. Note that cracks in alluvium are counted in a cumulative way, while cracks in mud are calculated per stage, due to the healing procedure for broken bonds in mud (cf. Sec. 2 and Al-Halbouni et al., 2018). These cracks cause, in addition to the porosity, further changes of the apparent shear modulus (Eqn. 6) and Poisson ratio (Eqn. 5) and hence reduce the effective apparent elastic modulus of the underground (Figure 9E). This is also expressed in the apparent shear wave velocity (Eqn. 3) for the same stage (Figure 9F). We observe strong changes in the central deep part of the model, where the largest void space growth rate exists. Remnants of earlier subrosion at shallower depths are nicely reflected in the apparent modulus and shear wave velocity distribution. More stable parts of the alluvium layers have higher values of E > 500 MPa and $v_S$ > 275 m/s (HVZ – high velocity zone). The lowest values of E < 100 MPa and $v_S$ < 100 m/s occur in the mud layer close to the zones with highest porosity and most cracks in the currently active area cavity growth (very low velocity zone – VLVZ). In between lies the low-velocity subrosion-affected part both in the mud and alluvium layers (low velocity zone – LVZ), corresponding to the areas of earlier cavity growth and overburden disruption. The model shows up to 75 % shear wave velocity reduction in the central

subrosion affected parts of the mud in comparison to the initial values, and up to 50 % for the alluvial overburden or contact zone between mud and alluvium.

## 4    Discussion

In this section, we discuss how realistic our numerical modelling results are in comparison to natural observations and what can be deduced in terms of process understanding. We first make some general points about the relationship between sinkholes and larger-scale depression is different karst settings. We then concentrate our comparison on results from remote sensing and geophysics for the very active sinkhole formation area at Ghor Al-Haditha at the Dead Sea. As a reminder, our model should be able to explain the following features of the karst landform evolution typical in that area (Figure 1Figure 2,cf. Al-Halbouni et al., 2018; Watson et al., 2018):

- In all materials, multiple sinkholes have formed with many clustered, coalesced and/or nested.
- Larger-scale depression zones with pronounced marginal cracks have also developed around the sinkholes and sinkhole clusters. Formation of sinkholes began before, or at the same time as, the appearance of the first marginal cracks of the depression zones. Lateral expansion of the depression occurs in tandem with sinkhole formation.
- Morphological differences depend in which material the sinkholes form: Low depth/diameter (De/Di) ratio for mud-flat sinkholes, high De/Di for alluvium sinkholes. Sinkholes in high strength materials have partly overhanging sides.
- The De/Di ratios of the larger-scale depressions are an order of magnitude lower than those of the sinkholes within them.

### 4.1    Implications for karst landforms of clustered sinkholes and large-scale depressions

As discussed by Ćalić (2011), for limestone karst areas, differences between enclosed depression types in karst regions occur in regard to scale, inter-relationship and morphometry. Sinkhole (or doline) diameters occur on a sub-100 m-scale, uvalas typically occur on a several hundred meter to km-scale in limestone karst, and so called poljes on even a larger scale. A single uvala typically includes numerous dolines within it, which led to the concept of uvala formation by doline coalescence (Gutiérrez et al., 2014; Waltham et al., 2005). The further development of dolines to uvalas and ultimately to poljes, is regarded by most workers as erroneous, and some do not consider uvalas to evolve by doline coalescence either (Ćalić, 2011). Although our simulations are purely mechanical and hence lack some important hydrological aspects for comparison to areas of limestone or evaporite karst, they nonetheless yield some new insights into the potential controlling factors on the inter-relationship between these different depression types.

Our models generally show that a differential subrosion pattern is necessary to achieve clustered sinkhole formation within a larger-scale depression. A spatially constant growth rate across the cavity array alone is not sufficient to generate sinkhole

clusters, even if the interacting cavities were at different depths. This is due to the resultant stress system, whereby a well-developed stress arch spans the entire constant-growth cavity array, which acts mechanically as a single entity. A differential cavity growth, on the other hand, produces more localised stress concentrations and arches above individual cavities in the array, leading to localised overburden failure and collapse into those cavities. The progressive deepening of subrosion is

particularly important to account for the observation at the Ghor al-Haditha site of initiation of larger-scale depression synchronously with or shortly after sinkhole development.

Our models also highlight conditions under which uvala-like depressions may or may not develop by sinkhole coalescence. In models with a relatively strong soluble layer, sinkhole coalescence is a mechanism for formation of a larger-scale uvala-like

depression (Figure 5D). This is because the relatively strong cavity-hosting layer inhibits deformation beyond the immediate cavity surroundings and promotes the formation of relatively large sinkholes that coalesce. In models with a relatively weak soluble layer, however, the uvala-like depression develops as a spatially- and temporally-distinct feature from the sinkholes within it. Rather than forming by sinkhole coalescence, the uvala-like depression reflects a wider-scale subsidence into the cavity array (or subrosion zone) as a whole that results from more widely distributed deformation in the weak cavity-hosting

layer (Figure 5A, B and C). The results of this latter model set-up are consistent with observations of the spatio-temporal relationships between sinkholes and uvalas in the evaporite karst examples at Ghor al-Haditha (Figure 2).

In our model results, we also reproduce the main relative morphometric attributes of sinkholes and uvala-like depressions as observed at the Ghor al-Haditha study area. As shown by Al-Halbouni et al. (2018), sinkhole depth/diameter ratios in the

models and nature are depend on overburden material properties, especially strength. Sinkholes in strong alluvium overburden have De/Di ~0.48 in the models compared with De/Di ~ 0.40 in nature; sinkholes in weak mud overburden have De/Di ~0.22 in the models compared with De/Di ~ 0.10 in nature (Watson et al., 2018). The uvala-like depressions at Ghor al-Haditha have a De/Di ~ 0.016 – 0.042 (Watson et al., 2018), which, as seen also for limestone karst settings (Ćalić, 2011), is about an order of magnitude less than the sinkholes. A similar relationship is seen in our models in which the larger-scale depressions have

De/Di~ 0.07-0.14; this ratio could have been made even lower simply by having a wider cavity array.

### 4.2    Detailed comparison with temporal development of subsidence at the Dead Sea sinkhole area

We analysed data from repeated photogrammetry of three consecutive years of the sinkhole area of Ghor Al-Haditha, at the eastern side of the Dead Sea (Figure 1A and Al-Halbouni et al., 2017). The datasets have been used to derive DSM difference maps between the consecutive years via GIS software.

Figure 10 shows the spatio-temporal evolution of recent sinkhole formations and patterns of holes, drainage channels, cracks and depression structures as observed in all cover materials in and around the main depression zone of the area (cf. Figure 1 and Figure 2). In the relatively strong alluvial sandy-gravel cover material (Figure 10A), we observe a cluster of rather deep and narrow sinkholes forming between 2014 and 2015. Small conical holes are precursors to the development of larger conical

sinkholes (I and II). A typical coalescence and partial overprinting of large and small holes can be seen at the lower right (III). The DSM difference in Figure 10A depicts the new sinkholes and lateral sinkhole growth. We observe a small overall subsidence between the new sinkholes, but a rather stable surrounding.

In the relatively weak clayey limestone carbonates material (Figure 10B), we observe the development of a cluster of typical wide and shallow sinkholes formed between 2015 and 2016. Similar to the alluvium, coalescence of individual holes into larger ones as well as the alignment of a series of different sized holes are observed. The development of new collapses during one year in this material can either show possible precursory structures (IV) or not (V). The scarps are generally not stable in time (VI) due to the weak material, as seen in the DSM difference map. An overall wider-scale subsidence of approximately 0.5 m +- 0.2 m is observed in the mud and between the sinkholes in the alluvium.

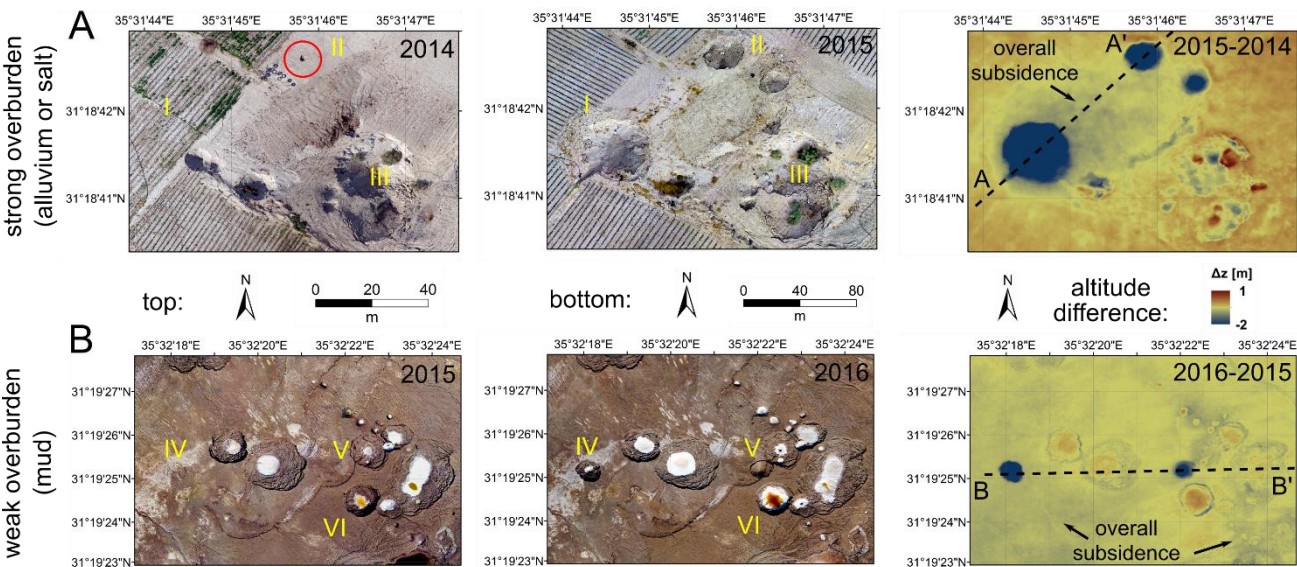

**Figure 10: Examples for subsidence and sinkhole formation at Ghor Al-Haditha from orthophotos and DSM difference maps. All orthophotos and DSMs have the same resolution (10 cm px$^{-1}$) and the accuracies are (horizontal, vertical): 2014 (10 cm, 11 cm), 2015 (12, 17 cm) and 2016 (37, 31 cm). (A) Vertical displacement between 2014 and 2015 in the alluvium. Large sinkholes have formed in the alluvium (I, II) with slight overall subsidence along cross-section A-A' from SW-NE. Red circle marks a small precursory hole at (II). Vegetation growth may cover subsidence effects in coalesced sinkholes (III). (B) Vertical displacement changes between 2015 and 2016 in the mud-flat. Precursory information (re-activated sinkholes) may exist (IV) or not be visible (V). Nests of sinkholes exist (VI) in an area of pronounced overall subsidence of 0.5 m +- 0.2 m determined for cross-section B-B' from W-E. Compare topographic profiles in Figure 11.**

For qualitative comparison with our models, Figure 11 shows the profiles across the DSM and vertical surface displacement for different stages of our models for weak and strong overburden. Although a precise matching is not intended, we clearly observe similar features in the modelled topographic profiles in comparison to the ones of the sinkholes/depression system at the Dead Sea. In weak material in the field, slight subsidence at the early stage is visible, revealing the contours of the future sinkhole (imprints), which were also observed in the model. In strong material, early collapse sinkholes may precursor further

large-scale collapses and nesting, both in nature and in the models. Sinkhole development is usually accompanied by fracturing at the margins of the larger-scale depression (see Figures 2, 5A, C and 11C). Large, deep fractures occur in strong material, while small, shallow fractures in the cohesive weak material (see also Holohan et al., 2011). In general, the fractures indicate a widening of the depression zone. Finally, because of the prescribed geometry of the subrosion zone, which is expected to be more complex in natural karst systems, and the limitation to 2D modelling, we cannot infer conclusions about the observed migration of such sinkhole clusters in nature (Figure 2).

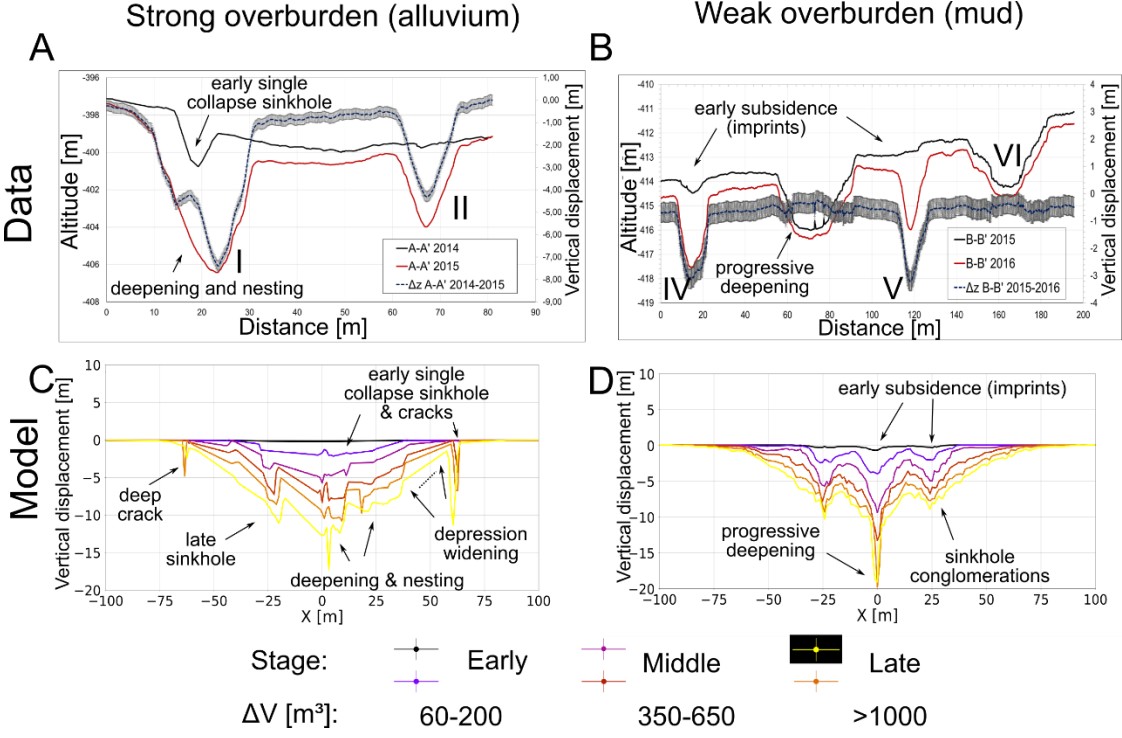

Figure 11: Topographic profiles data as indicated in Figure 10 and corresponding models. Top row: Topography and vertical displacement. Figure 10 of (A) cross-section A-A' from SW-NE in the alluvium and (B) cross-section B-B' from W-E in the mud-flat. Bottom row: Representative topographic profiles across final models for (C) high strength alluvium on mud and (D) low-strength lacustrine mud.

## 4.3 Subsurface patterns of sinkhole clusters and subrosion

From shear wave reflection seismics, zones with low reflectivity and velocity inversion anomalies in the S-wave velocity field are indicators for zones of material depletion or faults (Wadas et al., 2016). In the central part of the sinkhole-affected alluvial fan system at Ghor Al-Haditha, a deep-seated ($\geq$ 60 m depth) main subrosion zone based on the determined top of a lacustrine mud layer has been identified by comparison of shear wave reflection profiles with borehole logs (Polom et al., 2018). In several profiles of that work, shallower subrosion zones can also be identified and a general dip tendency of the deeper layers towards the NW is observed, indicating a Gilbert type alluvial fan foreset/topset system. We picked a section of profile one (cf. Figure 1) of Polom et al., (2018) as an example, and we present an interpreted version of the shallower part in Figure 12A.

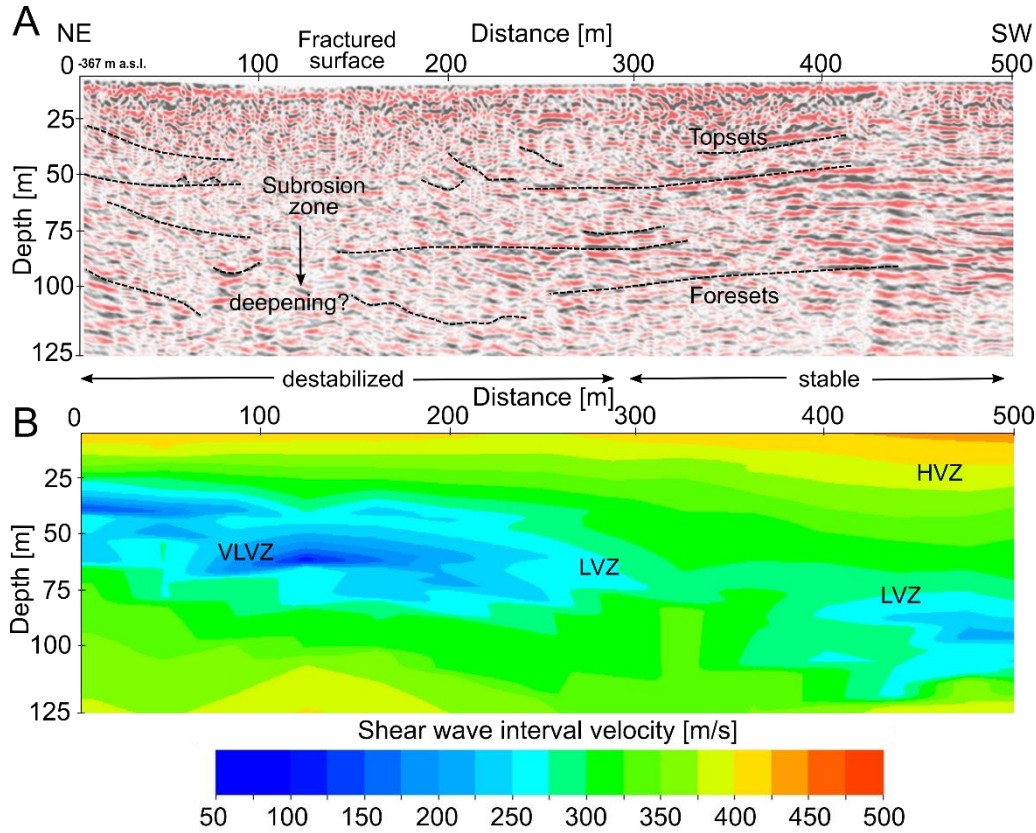

**Figure 12: Subrosion affected parts of shear wave reflection seismic profile 1 at the sinkhole site of Ghor Al-Haditha. (A) Modified and interpreted first 500 m of profile 1 after** *Polom et al.*, 2018**. (B) Shear wave interval velocity versus depth with marked very low (VLVZ), low (LVZ) and high velocity zones (HVZ).**

A layered system of alluvial fan sediments with stronger reflections can be seen to the SW, while the central and NE parts, close to the main depression zone, are affected by downsagging of up to several meters, by disturbed layers, and by bowl- or cone shaped features in the upper 50 m. Near surface uneven reflectors may indicate local fracturing of the layers. Locally, more stable parts, i.e. stronger reflections, exist. This is comparable to the subsurface structure as found in the final stages of the alluvium-on-mud model (Figure 5A). The stable blocks are especially clearly visible in the incremental strain evolution plots (cf. Figure 17 in Appendix A.4). An indication of a deepening subrosion zone can be inferred from the change in the transparency of the reflectors.

Figure 12B shows the 2D-field of shear wave interval velocities in depth of the same profile section. It was derived after Dix, (1955), based on the 2D Root Mean Square (RMS) mean velocity field in time resulting iteratively from interactive velocity analysis of the hyperbola moveouts for the Common Midpoint (CMP) stacking procedure, which was subsequently iteratively evaluated and optimized by migration velocity analysis. The velocity field reflects the general survey situation of a relatively high velocity of 400-425 m/s close to the surface caused by the road construction (asphalt surface over a compacted man-made

gravel infill) and reduced velocities of 300-375 m/s below for the natural alluvial sediments. The lateral structure mainly correlates with the structure image in Figure 12A. Low velocity values of 275 m/s down to ca. 100 m/s (light blue to blue zones) indicate either subsurface zones of true low velocities due the specific lithologic formation, i.e. soft sediments, or zones of disturbed formations where the shear modulus is reduced due to mechanical stress of the formation by disruptions, caused

by an upward propagating deformation process. Very low values below 100 m/s may indicate areas where the shear wave could not propagate the area by a straight ray path and turned around it, e.g. in case of a cavity ($G_{dyn} = 0$) or a collapsed zone of very low shear modulus. In this case, the resulting elongated propagating paths compared to the regular straight paths lead to zones of apparently very low interval velocities of less than 100 m/s, partly close to zero, which are not realistic for true lithologic units.

The decrease in apparent seismic shear wave velocity has been attributed by Polom et al., (2018) to diminished grain coupling (either by pore pressure effects or enhanced fracturing of the rocks) and to the influence of a high-velocity surface layer (e.g. asphalt). The simulated apparent velocity values of Sec. 3.2 lie in the range of the field estimates with a strong reduction in the simulated mud layer during cracking and collapse. We consider the presented simulation stage of Figure 9 as most appropriate to explain the observed shear wave velocity reduction, which we interpret to be caused by enhanced fracturing, i.e. crack

density increase, porosity increase and consequent modulus reduction in a deepening differential subrosion zone. As such, our final model qualitatively and quantitatively aids the interpretation of the subsurface geophysical patterns for the material combination as found at Ghor Al-Haditha field site.

A conceptual model in Figure 13 summarizes the main findings of this study and how they relate to the complex, dynamic karst

systems in nature. A large-scale depression builds up due to distributed material removal in the underground by subrosion in a karstic drainage network. Nested and/or clustered sinkholes may appear with relatively stable blocks in between. Lateral material heterogeneities may cause different sinkhole morphologies and surface expressions of cracks/fractures that surround the large-scale depression. Depending on the material strength, large-scale depressions may build up either by sagging, block-wise brittle failure, lateral widening or coalescence of sinkholes. The subsurface shows strong layering disturbances and

porosity and modulus changes leading to a low seismic velocity zone (LVZ). The pre-collapse principal stress system is divided into individual stress arches due to the differential subrosion pattern. Water infiltration generally may cause additional superficial dissolution structures.

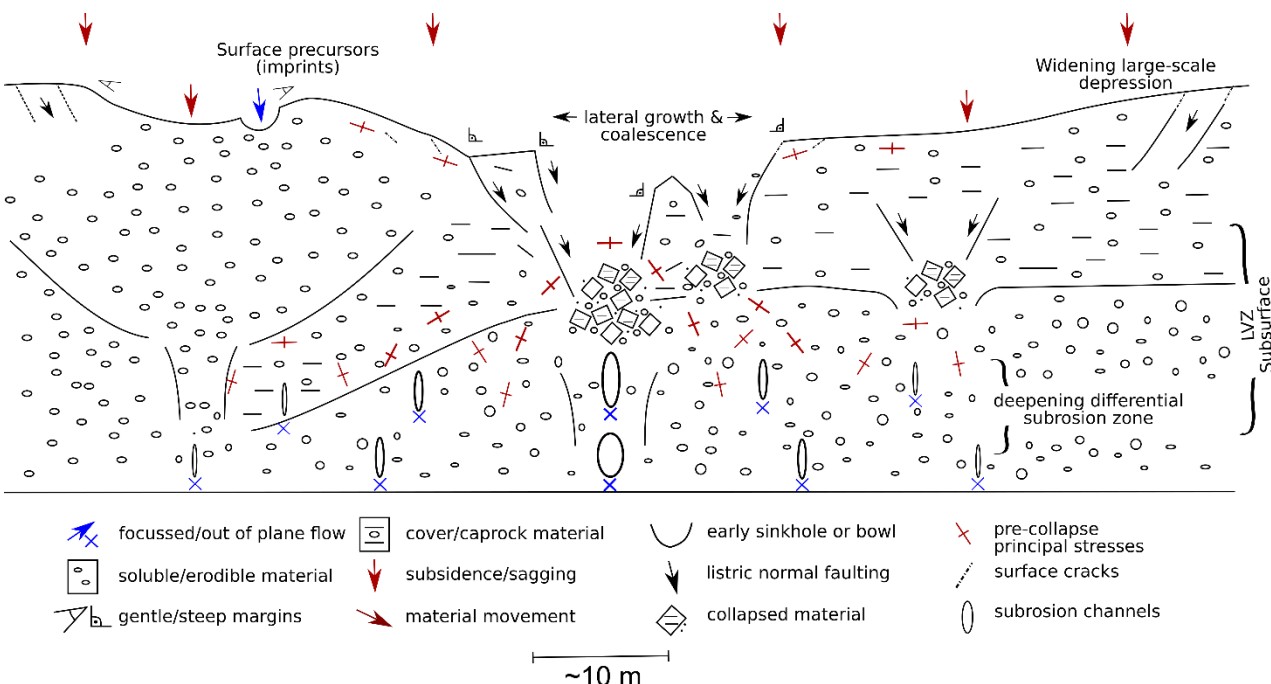

**Figure 13: Conceptual model of sinkhole cluster and large-scale depression development. Several sinkholes of different stages, types and varying subrosion depths are indicated in this sketch (centre - caprock sinkholes, left - suffosion sinkhole).**

## 5    Summary and conclusions

In this study, we presented a physically realistic 2D distinct element numerical modelling approach to simulate the growth of a system (array) of karstic cavities with the subsequent formation by subsidence of multiple (clustered) sinkholes within a larger-scale (uvala-like) depression. Two end-member growth scenarios of the multiple cavity array were tested with the following main outcomes:

1. Cavity growth at the same depth level and growth rate yields a stable compression arch around the entire cavity array. This scenario hinders individual sinkhole collapses but favours a simple block-wise subsidence spanning the whole cavity array.

2. Cavity growth at progressively deepening levels with varying growth rates is characterised by a heterogeneous, interacting stress pattern in the cavity array and overburden. This scenario favours the more complex formation by
subsidence of individual sinkholes and sinkhole clusters within a larger-scale, gentler (uvala-like) depression.

The influences of geomechanical variation in growth scenario (2) were further investigated by simulating four different layered combinations of low- and high strength materials representing the cavity-hosting medium and its overburden. The model

results were also compared with surface morphologies from remote sensing and with subsurface structures from geophysical studies at the active sinkhole formation area near Ghor Al-Haditha, at the Dead Sea. We found that:

- For models with a weak cavity-hosting material, cavities remain small throughout and wide-scale deformation in the cavity-hosting material and the overburden is promoted. This leads to development of a large-scale depression formed by subsidence that is structurally distinct from the individual sinkholes and is geometrically linked to the cavity array as a whole.

- For models with a strong cavity-hosting material, large cavities can develop before the overburden collapses into them and wider-scale deformation is inhibited. Consequently, a larger-scale depression forms in this case mainly by coalescence of sinkholes.

- Deepening of the differentially-growing cavity array in model scenario (2) leads to sinkholes forming synchronously with, or just before, the development of a larger-scale synclinal depression. This order of appearance of sinkholes relative to the larger-scale depression is observed at the Ghor al-Haditha sinkhole area. The modelling condition of deepening cavity growth is representative of a base-level fall, the main hydrogeological boundary condition ocurring at the Dead Sea shoreline.

- Morphometric relations (depths and diameters) for both sinkholes and large-scale depressions as observed in nature are successfully reproduced in the models.

- Subsurface structures and calculated shear wave velocities match to a high degree those inferred from field estimations in shear wave seismic data. A low seismic velocity zone (100-275 m/s) is imaged and simulated, compatible with the existence of a deepening subrosion zone at the Ghor al-Haditha field site.

Finally, we conclude that the presented numerical modelling approach of multiple cavity growth has proven to be successfully applicable to sinkhole-depression systems and that it provides a basis for enhanced geomechanical understanding of karst development and hazard assessment.

## Author contribuition

DAH and EPH led the production of figures and writing of the manuscript. DAH undertook the majority of the data analysis associated with the DSMs and Modelling. EPH, DAH, RAW, UP undertook the field studies and close-range photogrammetric surveys in 2014 – 2016. UP and RAW contributed with satellite image and seismic data analysis. All authors reviewed and commented on the manuscript, and they contributed to discussions of the data.

## Conflict of interest
The authors declare that they have no conflict of interest.

## Acknowledgements

We would like to acknowledge our colleagues from MEMR, namely Ali Sawarieh and Hussam Alrshdan, for their continuous support. Particular thanks go to Damien Closson, Thomas R. Walter, Marc K. Elmouttie and Charlotte M. Krawczyk. We would additionally like to thank the anonymous reviewers and the editor for the fruitful discussion. The authors acknowledge the financial support by the Helmholtz DESERVE Virtual Institute and the Federal Ministry of Education and Research of Germany in the framework of SIMULTAN (grant 03G0843). Finally, particular thanks go to Itasca for providing the license of PFC-V5 in the framework of the IEP and the German Academic Exchange Service (DAAD) for a short-term Doctorate research grant.

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

# Appendix A    Numerical simulation of multiple void spaces with DEM

## A.1 Cavity growth implementation

The cavity growth function f(i), which relates the initial removed area $A_0$ to the area increment to be removed $A_i$ (Al-Halbouni et al., 2018), has been updated to account for multiple voids that can start and stop growing at defined intervals ($i_0$, $i_{max}$). The function hence depends on each single void of index j and the formula becomes:

$$A_i \;=\; f_j(i,..)A_0, \qquad i \in [i_0, i_{max}]$$

The linear void space growth function relating initial void space area with the removed area at further intervals stands as an approximation for real fracture or void growth by physio-chemical processes in karst aquifers. Pure chemical dissolution of limestone or gypsum versus fracture widening shows a linear behaviour as long as the concentration of the undersaturated incoming fluid is lower than 90 % of the equilibrium concentration for that mineral (Dreybrodt et al., 2005; Kaufmann and Dreybrodt, 2007; Romanov et al., 2010).

## A.2 Optimal model development

For finding the optimal model, we generally define five individual semi-elliptical voids with a distance of 25-40 m from each other. They belong to one of three initial size groups and one of three void space growth function classes. Initial areas of set 1 (small) are $A_{0,1} = 2.7 - 6$ m³ and linear eccentricity $e_1 = 1.7 - 2.2$, of set 2 (mid-sized) $A_{0,2} = 10 - 14$ m³ and $e_2 = 3.5 - 4.0$ and of set 3 (big) $A_{0,3} = 24.5$ m³ and $e_3 = 5 - 5.5$  Material removal of set A (slow) has an incremental function of $f_{j=1}(i) = 1.0^i$, set B (mid-speed) $f_{j=2}(i) = 1.05 - 1.075^i$ and set C (fast) $f_{j=3}(i) = 1.1^i$. The subrosion zone is defined in different depth below the surface, set I (shallow) is for 20 m depth, set II (middle) for 30 m depth, set III (mid-deep) for 40 m depth, set IV (deep) stands for 50 m depth and set V (very deep) stands for 60 m depth. Representative for different material combinations all results of the tests are shown in Figure 14 for alluvium on mud layer setup. The following table summarizes the different tested void space setups. The results for the final model are shown in Sec. 3.

**Table 3: Tested sets of void spaces for the DEM models. All of them were applied to the material settings common at the Dead Sea shoreline. The model set combination is given in terminology: Initial area set/void space growth class/subrosion depth set. So, e.g. 1/A/I would stand for small sized, shallow seated void spaces growing at a constant rate.**

| Removal zone (void space)/ model set name | A: All voids the same $(A_{0,2})$ | B: Three inner voids shallower $(A_{0,2})$ | C: Two outer voids shallower $(A_{0,2})$ | D: Two outer voids accelerating growth $(A_{0,2})$ | E: All voids differential growth rate but same initial areas $(A_{0,2})$ | F: All voids same growth rate but variable initial areas $(A_{0,2}, A_{0,3})$ | G: All voids differential growth rate and variable initial areas $(A_{0,1}, A_{0,2}, A_{0,3})$ | H: Deepening differential growth rate and variable initial areas $(A_{0,1}, A_{0,2}, A_{0,3})$ | I: Final model: Deepening differential growth rate and variable initial areas $(A_{0,1}, A_{0,2}, A_{0,3})$ |
|---|---|---|---|---|---|---|---|---|---|
| **central** | 2/A/III | 2/A/II | 2/A/III | 2/A/III | 2/C/III | 3/A/III | 3/C/III | 1: 3/C/I 2: 3/C/III 3: 3/C/V | 1: 3/C/I 2: 3/C/II 3: 3/C/III 4: 3/C/IV |
| **two inner** | 2/A/III | 2/A/II | 2/A/III | 2/A/III | 2/B/III | 2/A/III | 2/B/III | 1: 2/B/I 2: 2/B/III 3: 2/B/V | 1: 2/B/I 2: 2/B/II 3: 2/B/III 4: 2/B/IV |
| **two outer** | 2/A/III | 2/A/III | 2/A/II | 2/B/III | 2/A/III | 1/A/III | 1/A/III | 1: 1/A/I 2: 1/A/III 3: 1/A/V | 1: 1/A/I 2: 1/A/II 3: 1/A/III 4: 1/A/IV |

**Constant void space growth (Test scenario 1)**: Figure 14A shows the evolution of a growing void space system of five voids of set A until surface collapse for multilayers of alluvium and mud. It shows cracks at the margins of the collapse zone and gradual sinking of a whole block. Individual, smaller scale sinkholes do not form. See main part of the manuscript for the description of the results for this test scenario.

**Constant void space growth with shallower inner voids**: Figure 14B shows the evolution of a growing void space system of five voids of set A with two inner voids 10 m higher than the others. It shows cracks at the margins of the collapse zone and gradual sinking of the whole block but with a division of the block into segments. Real individual smaller scale sinkholes do not form but are only an effect of the segmentation by the higher lying voids.

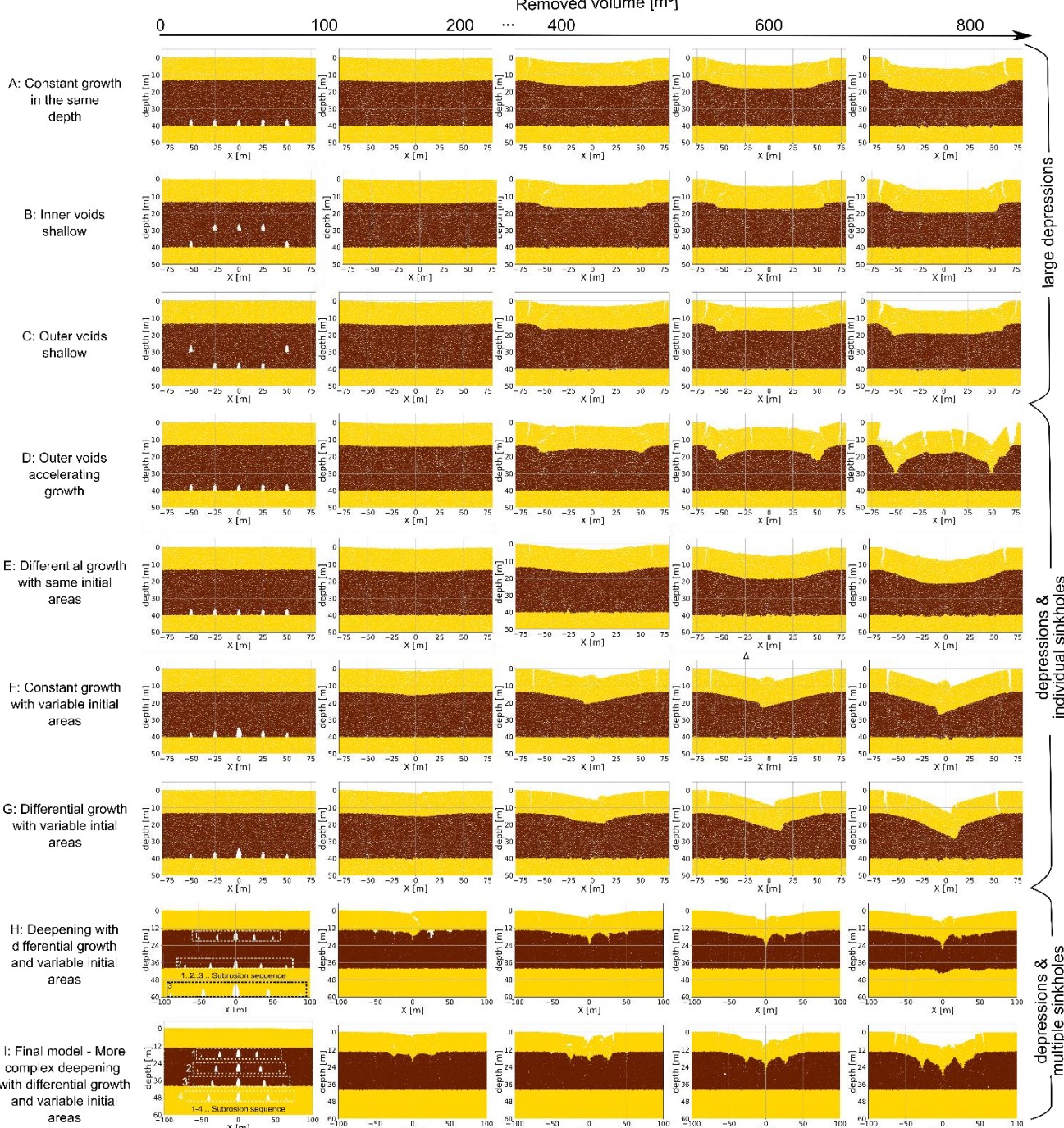

**Figure 14: The influence of void space geometry (sizes and positions) and material removal speed in an alluvium on mud layered system. These simulations (A)-(I) are essential to determine step by step the optimal model setup to achieve multiple sinkhole collapses in a large depression zone. Shown are only the core zones of the models at different stages of removed volume indicated above each plot. Note the slightly different size of plots (H, I) in order to account for the widening and deepening subrosion zone. For simplicity, no passive marker are applied in these images.**

**Constant void space growth with shallower outer voids:** Figure 14C shows the evolution of a growing void space system of five voids of set A with the two outer voids 10 m higher than the others. It shows cracks at the margins of the collapse zone and gradual sinking of the whole block and toppled blocks at the margins. Real individual, smaller scale sinkholes do not form, only a large and rather flat depression.

**Accelerating growth of outer voids:** Figure 14D shows the evolution of a growing void space system of three inner voids of set A with the two outer voids of set B, which leads effectively to an accelerated growth of the outer voids. We observe cracks and toppled blocks at the margins of the collapse zone, a gradual sinking of the whole block and first individual, but very large sinkholes. The deepest part of the depression is one of the sinkholes. The convex shaped bending of the middle part is not

observed in our field study.

**Differential growth with same initial areas:** Figure 14E shows the evolution of a growing void space system of the 2 outer voids of set A, the two inner voids of set B and the central void of set C, effectively an accelerating growth for the inner voids. We find cracks and toppled blocks at the margins of the collapse zone, a gradual sinking of the whole block. A large-scale

more steep sided depression forms.

**Constant growth with variable initial areas:** In Figure 14F the same effect as in the previous model can be produced by larger initial areas of the inner void spaces, with a largest material removal zone in the center. The growth rate is constant for each individual void space leading but the initial sizes differ. Here, we produce a compression ridge at the center of the

20 depression zone.

**Differential void space growth with variable initial areas:** In Figure 14G a combination of a larger starting area with a fastest growing rate in the center as in Figure 14E and F is used to achieve an accelerating differential void space growth. We can see the same effect as in the previous models but with a first formation of a small scale sinkhole in the center of the

25 depression.

**Deepening differential void space growth (Test scenario 2):** Figure 14H is a pre-final model accounting for the base-level fall affecting the subrosion zone depth. A combination of a larger starting area and deepening from levels I (20 m) to III (40 m) to V (60 m) is implemented, halting the previous subrosion when the new one is activated. We can see already complex

structure of individual, nested sinkholes in a large scale depression. This process is refined for the final model shown in Figure 14I, using a more complex combination and intermediate steps of subrosion zone deepening from levels I to IV. This leads to clearer development of multiple nested sinkholes that subside into a large depression zone the deeper the subrosion zone lies. See main part of the manuscript for a detailed description of the outcomes of this test scenario.

## A.3 Principal stresses in a multiple void space system

Figure 15 and Figure 16 show the developed maximum $\sigma_1$ and minimum and $\sigma_3$ compressive stress for the constant subrosion versus differential subrosion setups (scenarios 1 and 2) and two different material combinations. A large compression arch spanning the cavity array develops scenario (1) but is more fragmented in the scenario (2). For the weak overburden rather individual, stabilizing compressive arches build up in the strong interlayer and hardly translate upward. The minimum compressive stress plots for both setups show similar behaviour. Tensile stresses are recorded near the surface for strong overburden material. In contrast, the strong interlayer beneath the weak material leads to strong tensile stresses lined up at the edges of the cavities with spalling phenomena for both subrosion schemes. This line is centrally broken in the differential subrosion scheme. Shear stress observations are discussed in Section 3.1.

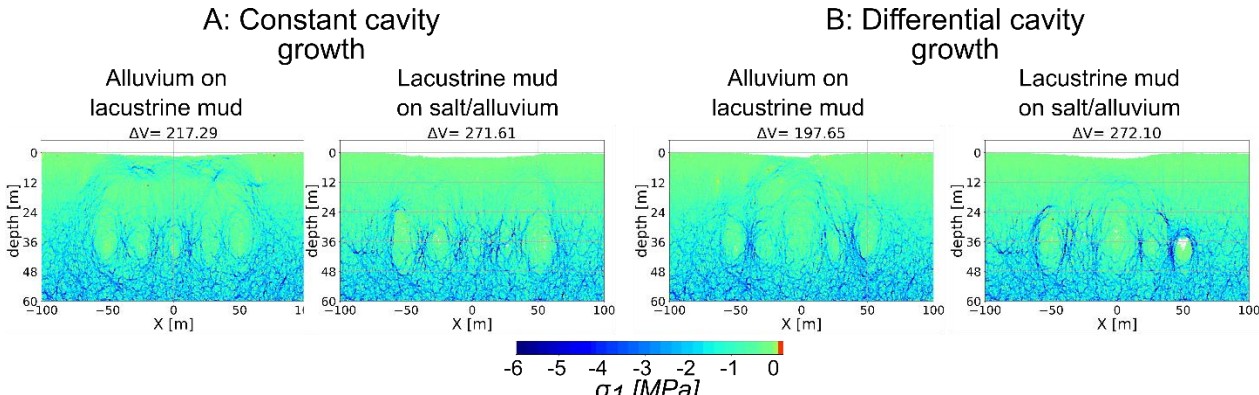

**Figure 15: Maximum principal stress around void spaces for (A) constant and (B) differential cavity growth scenarios models. Chosen are two material combinations where the subrosion affected layer differs in strength: alluvium on mud multilayer and mud on salt/alluvium succession. Shown are critical stages after void space installation followed by or exactly during overburden collapse for the same particle assembly. The removed volume [m³] is shown above the plots.**

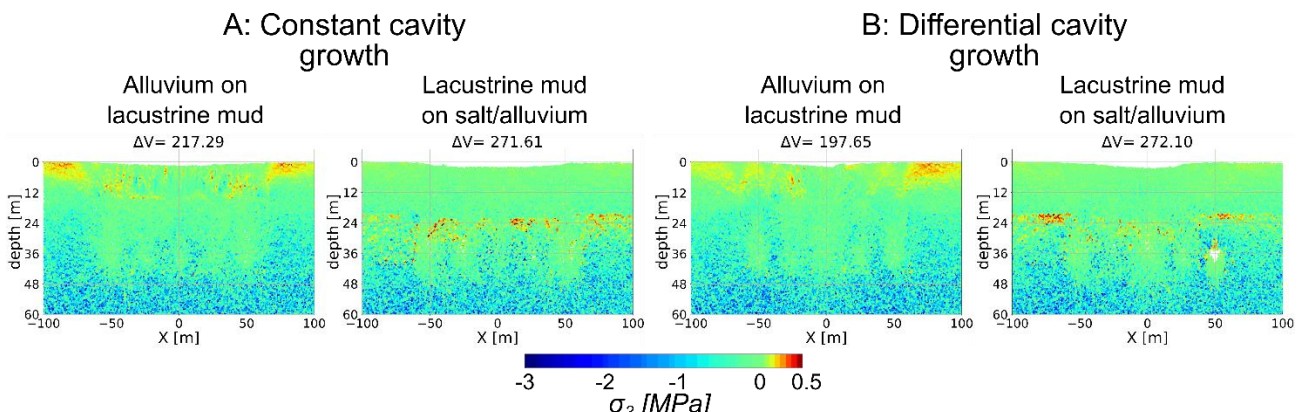

**Figure 16: Minimum principal stress around void spaces for (A) constant and (B) differential cavity growth scenarios models. Chosen are two material combinations where the subrosion affected layer differs in strength: alluvium on mud multilayer and mud on salt/alluvium succession. Shown are critical stages after void space installation followed by or exactly during overburden collapse for the same particle assembly. The removed volume [m³] is shown above the plots.**

## A.4 Incremental shear strain evolution

Figure 17 shows the incremental shear strain evolution for all simulated material combinations for the differential deepening subrosion scenario (2). The different mechanical response to material removal in the subsurface is nicely illustrated by these images. Strong strain localization occurs in all models in the material removal zones, at the boundaries of the depressions and at the margins of formed fractures. The continuous evolution of cracks into long fractures is nicely imaged as well.

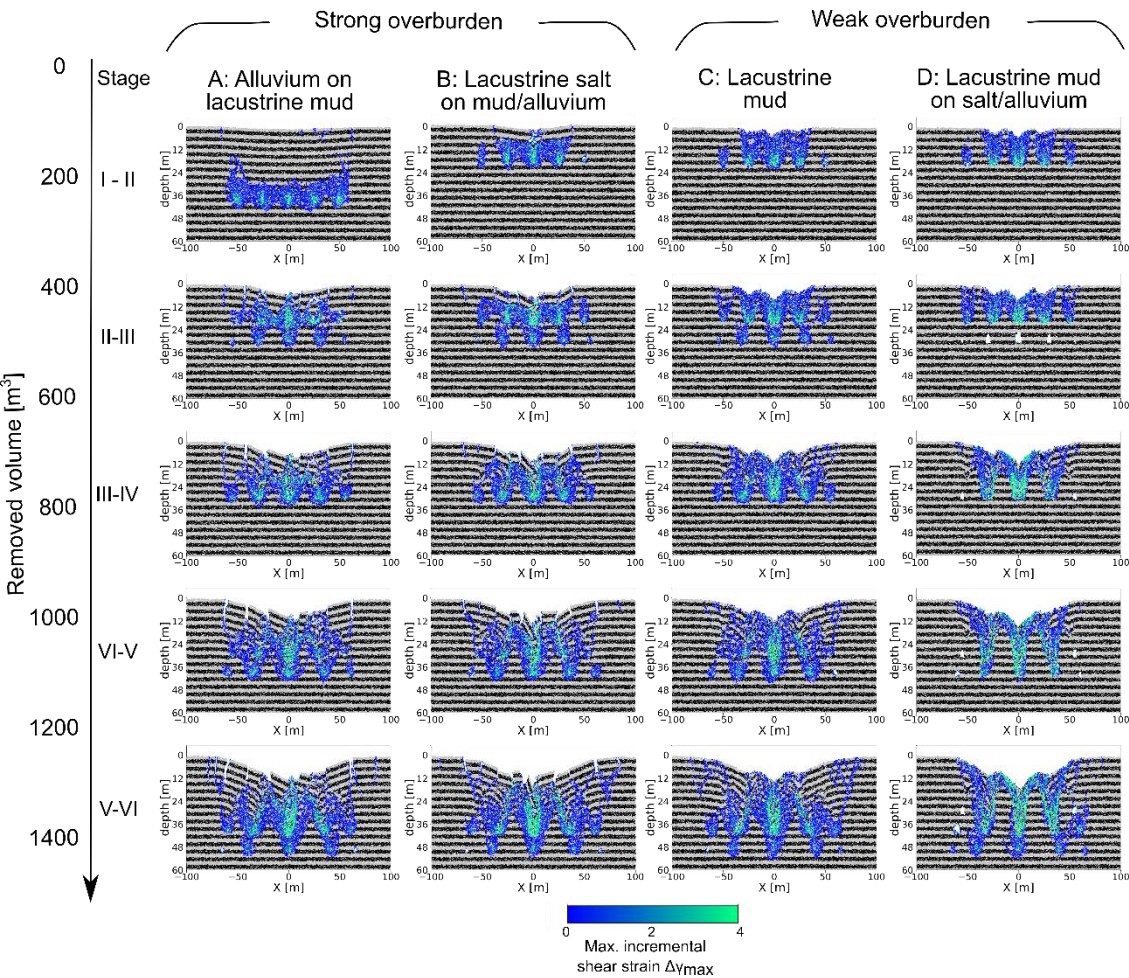

**Figure 17: Incremental shear strain evolution in between the simulation stages for four different material combinations common at the Dead Sea shoreline. Strong overburden: (A) alluvium/mud succession and (B) salt on mud/alluvium succession. Weak overburden: (C) pure lacustrine mud and (D) mud on salt/alluvium succession. The plots refer to the difference of maximum strains between two simulation stages indicated to the left. Refer to Figure 5 for individual stages. Note that passive marker layers are applied to highlight structural features.**

## A.5 Maximum shear stress evolution

Figure 17 shows the maximum shear stress evolution for all simulated material combinations for the differential deepening subrosion scenario (2). The stress is best imaged prior to collapse, i.e. the snapshots refer to instable moments except for the strong cavity hosting material. Localized and fragmented stress patterns can be observed in all models, with maxima for the mechanically strong overburden and cavity hosting materials (salt & alluvium) and delamination patterns due to the modulus contrasts.

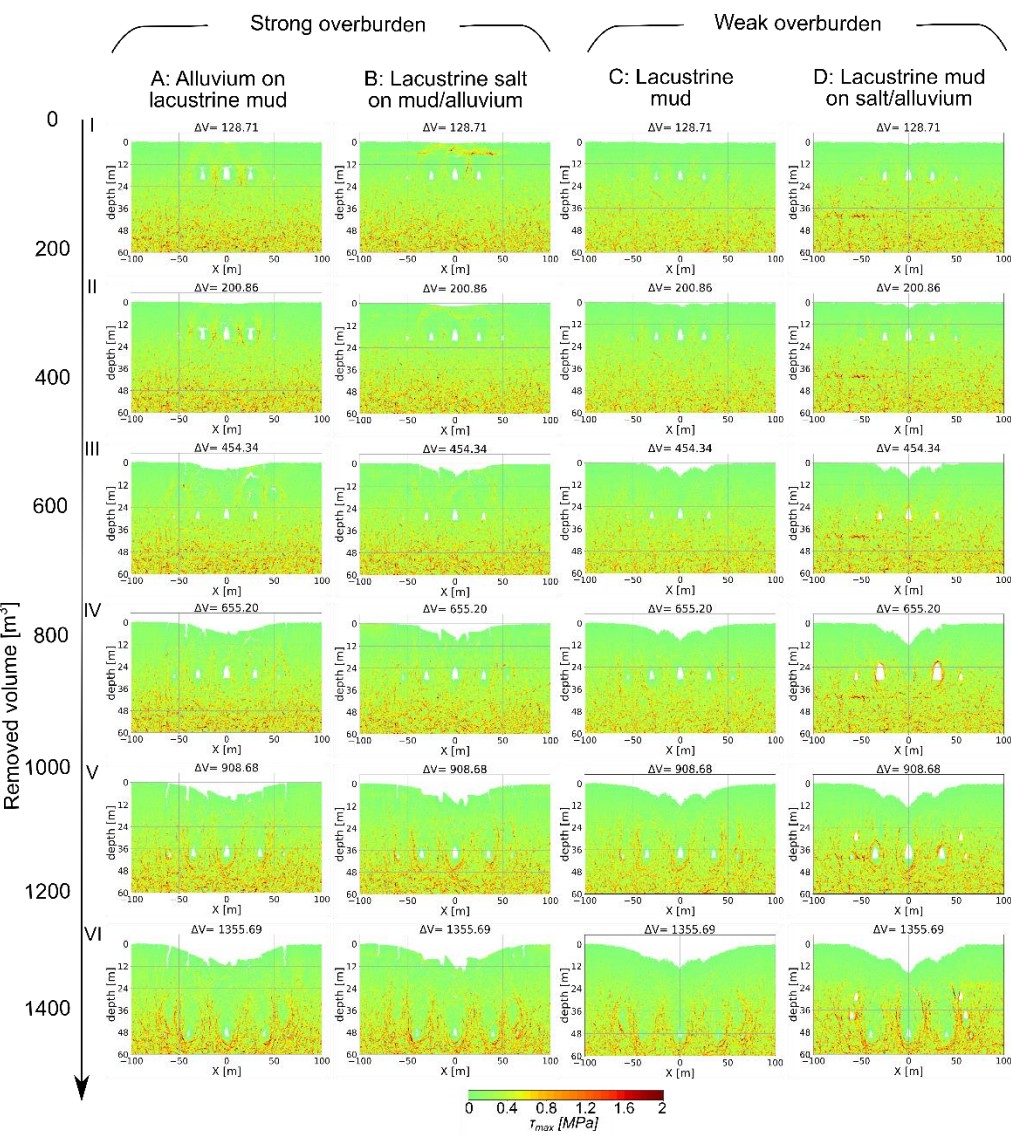

**Figure 18: Maximum shear stress evolution for four different material combinations common at the Dead Sea shoreline. Strong overburden: (A) alluvium/mud succession and (B) salt on mud/alluvium succession. Weak overburden: (C) pure lacustrine mud and (D) mud on salt/alluvium succession. Note that snapshots show instable stages before cavity collapse except partly for salt cavity hosting material.**