# Peer review of "Distinct Element geomechanical modelling of the formation of sinkhole clusters within large-scale karstic depressions"

_Solid Earth, 2019_

## Referee Comment (RC1) · Renaud Toussaint (Referee) · 5 Mar 2019

This manuscript proposes a parametric study of numerical simulations reproducing the formation of sinkholes in different types of settings in Karsts, and the discussion/comparison of the simulation results with real sinkhole fields observed in the Dead Sea region, depending on the formation modes and ground properties.

The article is well written, illustrative, the simulations are convincing, the discussion is interesting, and its conclusions are well supported by the material provided. The phenomenon is interesting for the readership of this journal, and I support its publication.

[Figure]

I have a few small questions/suggestions :

1. In the codes used, what is the dynamics of particles, i.e. the interaction laws adopted ? What is the influence of the viscosity during the shocks on the patterns observed? What is the influence of the inertia during the impacts ? Is the code overdamped ? Are the mechanical waves playing any role in the code dynamics ? Are the results dependent on the dissipation modes chosen (viscosity) ? Of the size of the particles ? How ?

2. Please precise the subrosion scenarios in more details – the first reading seems to show that particles are all removed at once, but subrosion rates are mentioned later on. Please provide a clearer / more detailed description. In general, there is an impact of removing particles in a single layer, or in several ones, as shown by the paper. This shows that the way of dissolving the particles mattters. Hence, some details could matter : What are the physical mechanisms controlling this subrosion rate ? Is it constant ? Is it homogeneous ? What if particles are removed progressively, as in a chemical dissolution process, with an expression linking kinetics and concentration in some species (and possibly, stress) ? (See e.g. : Koehn, D., Renard, F., Toussaint, R., & Passchier, C. W. (2007). Growth of stylolite teeth patterns depending on normal stress and finite compaction. Earth and Planetary Science Letters, 257(3-4), 582-595. Szymczak, P., & Ladd, A. J. C. (2004). Microscopic simulations of fracture dissolution. Geophysical research letters, 31(23). Toussaint, R., Aharonov, E., Koehn, D., Gratier, J. P., Ebner, M., Baud, P., ... & Renard, F. (2018). Stylolites: A review. Journal of Structural Geology. )

3. Is it possible to relate such dissolution kinetics to the motion of the waterbed, and the simulation of concentration fields associated to it ? What is the concentration profiles leading to the differential erosion rate ? Is there any influence of damage due to precipitation during drying, as e.g. in : Noiriel, C., Renard, F., Doan, M. L., & Gratier, J. P. (2010). Intense fracturing and fracture sealing induced by mineral growth in porous rocks. Chemical Geology, 269(3-4), 197-209. ?

4. P15, line 2 : " It must be noted that porosities over 0.5 have not been translated into apparent elastic moduli, the latter is assumed to be zero then. ": It would be better to use a different color there, since in reality even unconsolidated granular media, under gravity, will have a non zero velocity. The velocity is not modeled, and the velocity is low, but it is certainly not zero. Please precise it in the figure to remove the ambiguity.

Some few typos follow :

P11, line 10 : " This produces deeper and sinkholes in the later stages of the model evolution. " : no " and "

P13, line 10 : "These aspect are better understood " : aspects

---

## Referee Comment (RC2) · Nestor Cardozo (Referee) · 25 Apr 2019

I find the manuscript very interesting, well supported by extensive and integrated field work and numerical modelling (some published), and well written. As such, I consider the manuscript should be accepted after minor revision. These are my main comments:

1. Assembly calibration: Presumably the tests mentioned in Table 2 were run in a sub-sample of the assembly in Figure 3. Please state the size and number of particles in the samples used for assembly calibration.

2. Material behaviour: Salt may be affected by viscous forces. To what extent disregarding these forces in the models, may have affected the analysis? e.g. models B and D in Figures 4 and 5. For salt as a host of cavity-hosting material (Figure 5D), wouldn't viscous forces prevent the formation of strong, open cavities in mature stages of the model?

3. To what extent element/particle size may affect the estimation of the modified Poisson ratio and shear modulus? The two equations below line 25 in page 8 are very reasonable, but to what extent, for example, reducing the particle radii by half (increasing the number of particles twice) would affect the estimation of these modified values?

4. Figure 9 is a nice summary of the processes observed in the models. But wouldn't it be better to complement this figure with plots of total and incremental strain (e.g. shear strain) evolution in the models? These plots will clearly illustrate the total and active deformation at any stage, and in my opinion will truly show the evolution of the different mechanical components introduced in Figure 10. Incremental strain can further support the observations made in Figure 7 using shear stress.

5. Figure 12 can be taken forward by adding another column containing the modelled shear wave velocity (Figure 8D), and a forward seismic model (since the distribution of acoustic parameters in the model is known) of the DEM model. This will allow comparing actual to modelled seismic. It may be beyond the scope of the manuscript but it would certainly make the manuscript more appealing.

Below are minor comments:

Page 1: "a feedback loop": Unclear

Page 1: "same level" change to "same depth level"

Page 1: "individual growth rate": Unclear

Page 1: "stress interaction": Unclear

Page 2: "The relatively straightforward tectonic setting": Unclear

Page 3: Remove "compositions"

Page 5: "uniform distribution between a minimum and maximum particle radius in a 2D box": Unclear, can this be illustrated in a small figure/inset in figure 3?

Page 6: Change "same level" to "same depth level"

Page 6: "Same depth of 40 m": Why? This is explained latter in the Appendix but at this stage it is not clear why the cavities are placed close to the basement top.

Page 7: "bond-healing procedure" and "recombination behaviour": Unclear

Page 9: "Sec. 0" change to "Sec. 1"

Page 13: Define maximum shear stress in terms of principal stresses

Page 13: Change "constant subrosion" to "the constant subrosion"

Page 14: "All derived parameters must be regarded as apparent": Unclear

Figure 12: Please show the location of the line of section

Page 18: "observe similar features there": Unclear, what do you mean by "there"?

Page 17, line 26: Change "ae" to "a"

Page 17: "Stable immediate surrounding": Unclear

---

## Author Comment (AC1) · 24 May 2019

*Renaud Toussaint (Referee)*

*renaud.toussaint@unistra.fr*

*Dear René Tussaint, I would like to thank you very much on behalf of all co-authors for the review of the manuscript. We would like to address the comments and suggestions by this answer. Some parts/figures of the manuscript were rewritten/edited to improve the readability without changing the scientific content, these are highlighted in the re-submitted version. Please note that additional supplementary material (videos of multiple sinkhole collapse simulations) have been uploaded to the supplement.*

*With best regards on behalf of all co-authors,*

*Djamil Al-Halbouni*

**Answers to comments of reviewer no. 1**

*1) Dynamics of the particles: The DEM code uses an explicit time-stepping algorithm to calculate forces and translation/rotation of the non-deformable, interacting particles solving the respective Newton-Euler-Equations. The inertia is calculated via the mass (volume & density) and velocity, and an incremental force law has been used. Indeed, the code is slightly overdamped, using the standard value of 0.7 for damping. The purpose of the high damping is to ensure a quasi-static model behaviour. Viscous dissipation was not used (though this is possible within the PFC contact laws). Particle size affects the absolute values of the material strength and the variability of elastic parameters derived from simulated rock mechanics tests (cf. Potyondy and Cundall 2004), and it can have some influence on the macroscopic model behaviour. Relative to the scale of the system modelled, the values of particle radii chosen here minimise such discrepancies (cf. Appendix to Holohan et al., 2015). Further details of the dynamics can be found in an extensive study on benchmarking, particle and model size influence, and parameter calibration of this code, published in 2018 in the same journal by the same authors (and referred to in the main text of the present manuscript).*

*2) The precision of subrosion procedure: Indeed, the description in the submitted manuscript has led to the misunderstanding that particles are all removed at once. Rather, they are removed progressively (incremental, refer to p 6 l 10) and the speed of removal ($f=1^i-1.1^i$)*$A_0$) has been stated in the appendix A1. This speed of removal is sufficiently slow and the increments sufficiently spaced in time that model can re-equilibrate to maintain quasi-static conditions. This part of the text is now updated (p 6), and a more precise formulation has*

*been used. The same procedure as for individual cavity formation has been used as investigated in the partner manuscript (Al-Halbouni et al. 2018, Solid Earth). Regarding the dissolution kinetics, this is definitely a subject to include and a relation of particle removal speed to concentration etc. is important. However, in our purely mechanical, quasi-static simulations we rely on a simplified, time-independent approximation of the process by progressively but incrementally removing the particles within small domains of the model (the cavity domains). Nevertheless, future simulations should include groundwater flow aspects and ideally dissolution kinetics of rock salt.*

*3) Concentration fields: As stated above, the simulations are purely mechanical, just mimicking the hydraulic process with a particle removal scheme. Please refer to page 6. So no concentration can be given as we do not simulate water and the chemistry of water. However, with coupled FEM-DEM modelling this would be an interesting approach for more realistic simulations in future.*

*4) Velocity in unconsolidated media: Seismic velocities of unconsolidated media are non-zero, that's true. But here we have to define an arbitrary threshold of porosity, above which the area is considered as empty, i.e. no particles or only a few, non-bonded ones at the bottom of a cavity. A shear wave velocity in air is zero, therefore we keep the colorscale in Fig.8. The threshold has been defined by looking at material calibration via uniaxial stress tests, for porosities above 0.5 those tests are not meaningful anymore, the particles totally disconnected and we consider the area as "empty".*

*Erroneous points in the manuscript: These typos, reference errors and mistakes have been corrected.*

*Additional changes:*

*Supplementary material of collapse videos of multiple sinkholes for different material assemblies has been uploaded.*

---

## Author Comment (AC2) · 24 May 2019

*Nestor Cardozo (Referee)*

*Nestor.cardozo@uis.no*

*Dear Nestor Cardozo, I would like to thank you very much on behalf of all co-authors for doing the review of the manuscript. We would like to address the comments and suggestions by this answer. Some parts/figures of the manuscript were rewritten/edited to improve the readability without changing the scientific content, these are highlighted in the re-submitted version. Please note that additional supplementary material (videos of multiple sinkhole collapse simulations) have been uploaded to the supplement.*

*With best regards on behalf of all co-authors,*

*Djamil Al-Halbouni*

**Answers to comments of reviewer no. 2**

*1) Assembly calibration: The tests were indeed run on subsamples of the assembly for each material. We added the requested information about sample size in the revised sec. 2.3. As noted on Line 26-27 of the revised manuscript, the calibration and benchmarking of the sinkhole modelling approach has been primarily part of the partner manuscript published in SE (Al-Halbouni et al. 2018). To avoid repetition, we just present a synthesis of that calibration work here.*

*2) Material behaviour: This is a correct observation to be considered. Salt can behave viscously even at low confining pressures (close to the surface) and perform creep at low deformation rates. However, several factors led us to restrain from going further into details on complex salt rock behaviour in this publication. The material at the Dead Sea consists of several evaporite deposits, e.g. aragonite, halite, gypsum and calcite. Including this would need a specific study on different evaporite rocks at varying differential stresses. In our case, we opted for halite as a representative, and at low temperatures and confining pressures with large deformation, cataclastic material behaviour with microcracking dominates (see e.g. Jackson & Hudec, 2017, Salt tectonics), which is well represented in the rock mechanical tests performed on the subsamples (see partner paper Al-Halbouni et al. 2018, figure 6). In our models we could capture the elasto-plastic behaviour of salt, and also the brittle-ductile transition. However, we do not consider time dependent (creep) behaviour in our quasi-static approach. This is a limitation of the DEM models with the chosen bond scheme, but, it is also clearly beyond the scope of this work and goes along with a more thorough investigation of several types of evaporite rocks with more complex material behaviour.*

*3) Particle size (and model size, i.e. resolution) indeed affects the estimation of rock mechanical parameters. Several authors have studied this for the parallel bond contact*

*scheme, e.g. Potyondy and Cundall 2004 and Martin Schöpfer (e.g. 2007,2009,2017). For the sinkhole models, we did not want to add upon these studies; instead we did an intensive study on particle and model size effects on the vertical surface displacement above a circular cavity, in comparison with analytical solutions. We came up with a trade-off solution between sufficient resolution & sufficient accuracy vs. computation time. We then used this "optimal" particle size distribution to determine rock mechanical parameters on subsamples with the same particle radii. This can be found in the partner paper published in SE in 2018, Appendix B. However, to answer the second part of the question, for the above reasons, we didn't take a survey on how the rock parameters change when, e.g. taking half the particle size, for this study. However, Holohan et al 2011 did test for mechanical effect of reducing the particle sizes by half and found little difference in the elastic parameters. This is in line with earlier result of Potyondy and Cundall (2004).*

*4) Strain evolution: Indeed the maximum shear strain and incremental shear strain as well as the maximum shear stress nicely complement the interpretation of the mechanical process leading to sinkhole/large depression formation. We included three new figures and texts after this suggestion. Now Fig. 8 shows the maximum strain for the same situations like Fig. 7 (Sec. 3.2.). The incremental strain evolution & maximum shear stress have been added to the appendix A4 and A5 to not overload the first part with figures. Nevertheless, they nicely show the different strain/stress distribution for several material combinations for the final models, and e.g. also nicely reveals crack/fracture patterns and remnant stable parts. We would like to thank you for this very useful suggestion.*

*5) Seismic: We restrained from adding another figure of forward calculation seismic model. We consider it somehow as repetitive, as we use seismic to derive model parameters (shear modulus, density), apply our material removal approach, and derive shear-wave velocity distribution. If we would then include a new forward model from the shear-wave velocity distribution it would form somehow a loop, which we did not find very appealing to the reader. Also, correctly, it would be better part of a manuscript specifically dealing with seismic in the field area, which is under preparation.*

*Erroneous points in the manuscript: Thanks to the reviewing effort, the typos, references, unclear formulations and mistakes have been corrected.*

*Additional changes:*

*Supplementary material of collapse videos of multiple sinkholes for different material assemblies has been uploaded.*

---

## Author Response (AR2)

**Information on "Distinct Element geomechanical modelling of the formation of sinkhole cluster within large-scale karstic depressions" by Djamil Al-Halbouni et al.**

Dear editor,

In the latest version we made some small final changes to improve the readability of the document (i.e. references, missing punctuation, word change). The scientific content has not changed. Please find the marked document appended.

With best regards

Djamil Al-Halbouni

[revised manuscript text omitted]